# No Free Lunch Theorem and Black-Box Complexity Analysis for Adversarial Optimisation

**Per Kristian Lehre**
School of Computer Science
University of Birmingham
Birmingham, United Kingdom
`p.k.lehre@bham.ac.uk`

**Shishen Lin**[*]
School of Computer Science
University of Birmingham
Birmingham, United Kingdom
`sxl1242@student.bham.ac.uk`

## Abstract

Black-box optimisation is one of the important areas in optimisation. The original No Free Lunch (NFL) theorems highlight the limitations of traditional black-box optimisation and learning algorithms, serving as a theoretical foundation for traditional optimisation. No Free Lunch Analysis in adversarial (also called maximin) optimisation is a long-standing problem [45, 46]. This paper first rigorously proves a (NFL) Theorem for general black-box adversarial optimisation when considering Pure Strategy Nash Equilibrium (NE) as the solution concept. We emphasise the solution concept (i.e. define the optimality in adversarial optimisation) as the key in our NFL theorem. In particular, if Nash Equilibrium is considered as the solution concept and the cost of the algorithm is measured in terms of the number of columns and rows queried in the payoff matrix, then the average performance of all black-box adversarial optimisation algorithms is the same. Moreover, we first introduce black-box complexity to analyse the black-box adversarial optimisation algorithm. We employ Yao's Principle and our new NFL Theorem to provide general lower bounds for the query complexity of finding a Nash Equilibrium in adversarial optimisation. Finally, we illustrate the practical ramifications of our results on simple two-player zero-sum games. More specifically, no black-box optimisation algorithm for finding the unique Nash equilibrium in two-player zero-sum games can exceed logarithmic complexity relative to search space size. Meanwhile, no black-box algorithm can solve any bimatrix game with unique NE with fewer than a linear number of queries in the size of the payoff matrix.

## 1 Introduction

### 1.1 Black-Box Optimisation and the No Free Lunch Theorem

Black-Box Optimisation (BBO) is crucial for optimising complex, unknown, or expensive-to-evaluate functions in real-world scenarios, such as aerodynamic design and hyperparameter tuning, where only input-output observations are available. Formally, BBO is the task of optimising objective functions from some function class $\mathcal{F}$ where $\mathcal{F}$ consists of $f : \mathcal{X} \to \mathbb{R}$, where the algorithm is limited to making queries to $f$ [11, 14, 36]. In this case, the algorithm is only able to sample and query the function value $f(x)$ of search points $x \in \mathcal{X}$ from a "black-box" or "oracle" without access to any description of the objective functions $f$. A similar framework can be extended to game-theoretic BBO, namely adversarial BBO. Specifically, adversarial BBO is the task that optimising payoff functions from some payoff function class $\mathcal{G}$ where $\mathcal{G}$ consists of black-box functions $g : \mathcal{X} \times \mathcal{Y} \to \mathbb{R}$, with a limited budget of function evaluations. Additionally, two players $x \in \mathcal{X}$ and $y \in \mathcal{Y}$ form

---

[*]Authors are listed in alphabetical order.

38th Conference on Neural Information Processing Systems (NeurIPS 2024).

a maximin optimisation (as presented in Figure 1). The original No Free Lunch Theorems for traditional optimisation can be summarised as follows:

> "For all possible metrics, no search algorithm or supervised learning algorithm is better than another when its performance is averaged over all possible problem instances [45, 44]."

Figure 1: Comparison between traditional black-box optimisation and maximin black-box optimisation. Instead of querying at $x$ in traditional optimisation, maximin optimisation queries at $(x, y)$ include both strategy $x$ and the best response $y$ from the opponent, i.e. $\max_{x \in \mathcal{X}} \min_{y \in \mathcal{Y}} g(x, y)$. Their interaction is converted to the payoff $g(x, y)$ in the given black-box model.

Wolpert and Macready [45] and Wolpert [44] have revealed the underlying facts about the usefulness of traditional black-box optimisation algorithms, including various randomised search heuristics (such as evolutionary algorithms and simulated annealing) and machine learning algorithms (such as supervised learning). In particular, it shows that the performance of all black-box optimisation algorithms [45] and learning algorithms [44], when averaged over all problem instances, is the same for any maximisation or minimisation tasks. Droste et al. [11] has provided a generalised NFL theorem with more realistic scenarios by relaxing NFL theorem holds from all problem instances to problem instances closed under permutation. Another seminal work by Schaffer [37] also showed a conservation law for the generalised performance of learning algorithms in classification problems.

Adversarial optimisation tasks, such as maximin optimisation, are more complex and often counter-intuitive compared to traditional optimisation problems. In adversarial settings, defining solution concepts, or establishing what is meant by optimality, is essential. Common solution concepts include 'Maximisation Over All Test Cases', 'maximin', and Nash Equilibrium. A 'free lunch' in adversarial optimisation implies that, for a given solution concept, some algorithms consistently outperform others when averaged across all possible problem instances. This phenomenon was demonstrated by Wolpert and Macready [46] for adversarial optimisation with respect to the 'maximin' solution concept (Definition 10). Similarly, Service and Tauritz [39] established a 'free lunch' result for the 'Maximisation Over All Test Cases' concept (Definition 9).

Beyond 'maximin' and 'Maximisation Over All Test Cases', Nash Equilibrium is another widely studied solution concept in adversarial learning and maximin optimisation. This concept is central to various applications, including adversarial learning models such as GANs [15, 20] and spatial games [19, 26, 32]. While previous studies by Wolpert and Macready [46] and Service and Tauritz [39] have demonstrated the existence of 'free lunches' in adversarial optimisation, the following key questions remain open:

(1) Does adversarial optimisation exhibit a 'free lunch' for all possible solution concepts?

(2) If we use Nash Equilibrium as the solution concept, how can we characterise the difficulty of black-box adversarial optimisation problems for problem-independent, possibly randomised, search heuristics?

In this paper, we answer Question (1) in the negative by showing a No Free Lunch theorem for Nash Equilibrium solution concept and address Question (2) by introducing black-box complexity tools.

## 1.2 Challenges and Technical Overview of the NFL and BBC Results

There are several challenges in showing the NFL and BBC results. First, we highlight the challenges and our technical details in the derivation process compared to previous NFL work. The classical No Free Lunch theorem applies proof by induction with respect to the size of the function domain. A natural idea is to extend the previous proof-by-induction method in our NFL proof. However, one of the most challenging aspects of deriving NFL for Nash Equilibrium (NE) is that the adversarial

setting significantly increases the difficulty since the problem depends on the performances of both the player and the opponents. To address this, we introduce two technical lemmas from game theory (i.e. Lemma 3 and Lemma 4) and help to construct the isomorphism between two different problem classes (i.e. Corollary 1 and Lemma 2), which is an essential step in the proof of the original NFL theorem for traditional optimisation (see Section B for more details). Another significant challenge is that we start with a very weak assumption: we only allow access to the payoff function and make no assumptions about its properties, such as convexity, continuity, or differentiability. In fact, no gradients, continuity, or differentiability make the NFL and black box results easier to prove (i.e., the function class lacks a specific structure). On the other hand, it makes it more difficult for the algorithm. Consequently, we have limited analytical tools to proceed. To analyse the black-box complexity of adversarial optimisation, we introduce Yao's minimax principle and apply our new NFL result. Please consult the proof of Theorem 4.2 for more details.

## 1.3 Contribution

We prove a new impossibility result on black-box adversarial optimisation. In a two-player zero-sum game setting, there is no free lunch with respect to an approximation of the real cost model when regarding the unique Nash Equilibrium as the solution concept. In other words, all black-box adversarial optimisation algorithms have the same expected performance over a uniform distribution of all possible problem instances with the unique Nash Equilibrium as the solution concept. It is the first step to resolve this long-standing open research problem about NFL in general black-box adversarial optimisation since [45, 46]. Our results highlight the significance of the choice of solution concepts and the limitation of general black-box adversarial optimisation. Additionally, we introduce the first general black-box model and the notion of black-box complexity for adversarial optimisation. Under this general black-box model, we provide the general lower bounds for query complexity of computing Nash Equilibrium in adversarial optimisation. Finally, we illustrate our results on examples of computing unique Nash Equilibrium in two-player zero-sum games.

## 2 Preliminaries

### 2.1 Notation

Given $n \in \mathbb{N}$, $[n] := \{1, 2, \cdots, n\}$. Given a finite set $\mathcal{X}$, we denote the permutation group by $\mathfrak{S}(\mathcal{X}) := \{\sigma : \mathcal{X} \to \mathcal{X} \mid \sigma \text{ is a permutation of } \mathcal{X}\}$. For any $v \in \mathbb{R}^n$, let $\text{SUPP}(v) := \{i \in [n] \mid v_i \neq 0\}$. We refer to query complexity or runtime as the number of payoff function evaluations until the given algorithm finds the optimum. We consider the search space $\mathcal{X} \times \mathcal{Y}$, where $\mathcal{X}$ and $\mathcal{Y}$ are finite and denote the cardinality of a set by $|\mathcal{X}|$. For any bitstring $z$, $|z|_1$ denotes the number of 1-bits in $z$. $X \simeq Y$ denotes the isomorphism between $X$ and $Y$. $X \simeq Y$ if there exists a one-to-one correspondence map from $X$ to $Y$.

Let $f : X \to Y$ be a function from a set $X$ to a set $Y$. If $A$ is a subset of $X$, then the restriction of $f$ to $A$ is the function: $f|_A : A \to Y; x \mapsto f(x)$. Let $f : X \to Y$ be any function and $A$ and $B$ be sets such that $X \subseteq A$ and $Y \subseteq B$. An extension of $f$ to $A$ is a function $g : A \to B$ such that $f(x) = g(x)$ for all $x \in X$. Alternatively, $g$ is an extension of $f$ to $A$ if $f$ is a restriction of $g$ to $X$.

In this paper, we assume that the black-box optimisation algorithms do not make the same query twice. This can be achieved by memorising the outcome of previous queries.

### 2.2 Solution Concepts

Solution concepts for classical function optimisation are not directly applicable to adversarial optimisation in general-sum or zero-sum game settings [33, 24, 35]. Each agent or player's payoff depends on not only its action but also the response from its opponents, and thus, we need to introduce different solution concepts to specify what kind of optimum we look for.

Pure Strategy Nash equilibrium is considered as our solution concept in this work. We are interested in whether the black-box adversarial optimisation can efficiently find any given game's Nash equilibrium. We use the formulation in [30] to define Nash equilibrium rigorously. In this paper, we only focus on Pure Strategy Nash Equilibrium (abbrev. NE).

**Definition 1.** (Nash) Consider a two-player game. Given strategy spaces $\mathcal{X} \times \mathcal{Y}$ and payoff functions $g, h : \mathcal{X} \times \mathcal{Y} \to O$, where $O$ is an ordered set which determines the outcome of candidate solution $\mathcal{X}$ on test $\mathcal{Y}$, $(x^*, y^*)$ is a Nash equilibrium if for all $(x, y) \in \mathcal{X} \times \mathcal{Y}$, $g(x^*, y^*) \geq g(x, y^*)$ and $h(x^*, y^*) \geq h(x^*, y)$. In particular, if the two-player game is zero-sum (that is, if $g(x, y) + h(x, y) = 0$ for all $(x, y) \in \mathcal{X} \times \mathcal{Y}$), then $(x^*, y^*)$ is a Nash equilibrium if for all $(x, y) \in \mathcal{X} \times \mathcal{Y}$, $g(x, y^*) \leq g(x^*, y^*) \leq g(x^*, y)$. The solution concept is defined by

$$S = \{(x^*, y^*) \in \mathcal{X} \times \mathcal{Y} \mid \forall (x, y) \in \mathcal{X} \times \mathcal{Y}, g(x, y^*) \leq g(x^*, y^*) \leq g(x^*, y)\}.$$

We also defer other solution concepts for comparison in the appendix. We use the formulation in [46, 39].

## 3 No Free Lunch Theorem for Computing Nash Equilibrium in Adversarial Optimisation

In this section, we prove a No Free Lunch Theorem for black-box adversarial optimisation. As a first step toward the No Free Lunch Theorem, we consider two-player zero-sum games with the unique NE as the same setting in [34], but we relax the restriction from potential games to general two-player zero-sum games. The original NFL theorem for traditional optimisation assumed all problems instances [45], and later, sharpened work also proved that this holds for functions 'closed under permutation' [38, 11]. We now explain what 'closed under permutation' means and how it can be extended to games.

Permutation closure of a set of functions means that let $\mathcal{X}, \mathcal{Y}$ be two finite sets and $f : \mathcal{X} \to \mathcal{Y}$ defined by $f(x_i) = y_i$. Let $\sigma$ be a permutation $\sigma : \mathcal{X} \to \mathcal{X}$ and we can permute the function: $f(\sigma(x)) := \sigma \circ f(x)$. Schumacher et al. [38] and Droste et al. [11] defined 'closed under permutation (c.u.p.)' with respect to a single search space $\mathcal{X}$. A class of functions $\mathcal{F} = f : \mathcal{X} \times \mathcal{Y} \to \mathbb{R}$ is called c.u.p. if for all $f \in \mathcal{F}$ and all permutations $\sigma \in \mathfrak{S}(\mathcal{X})$, $f \circ \sigma \in \mathcal{F}$. For our adversarial setting, we need to extend this notion to $\mathcal{X} \times \mathcal{Y}$.

**Definition 2.** Given two-player zero-sum games, suppose $\mathcal{F}$ as a subset of all the payoff functions in these games $g : \mathcal{X} \times \mathcal{Y} \to O$ where $O \subseteq \mathbb{R}$. We say $\mathcal{F}$ is c.u.p if for any $g \in \mathcal{F}$ and any permutations on $\mathcal{X}, \mathcal{Y}$ denoted by $\sigma \in \mathfrak{S}(\mathcal{X}), \tau \in \mathfrak{S}(\mathcal{Y})$, we have $(\sigma \otimes \tau) \circ g \in \mathcal{F}$ (or abbreviated $(\sigma\tau)g$) where $(\sigma \otimes \tau) \circ g(x, y) := g(\sigma(x), \tau(y))$ for all $(x, y) \in \mathcal{X} \times \mathcal{Y}$.

If a set of two-player zero-sum games is c.u.p., we will also call it structure-free. In this paper, we restrict to the case that $\mathcal{X}, \mathcal{Y}, O$ are finite sets following the settings in [45, 46]. The proof of the No Free Lunch theorem for traditional optimisation by Droste et al. [12] is by induction, where one assumes that the statement is true for all "smaller problems". Here, a smaller problem refers to the following definition of a sub-game or a sub-problem class. Next, we define a sub-problem class for two-player zero-sum games.

**Definition 3.** Given two-player zero-sum games, let $O \subseteq \mathbb{R}$ and $\mathcal{F}$ be any subset of the set of payoff functions in these games $g : \mathcal{X} \times \mathcal{Y} \to O$ such that $\mathcal{F}$ is closed under permutation. For any given $(x_1, y_1) \in \mathcal{X} \times \mathcal{Y}$, any function $b_1 : \mathcal{Y} \to O$, and any function $b_2 : \mathcal{X} \to O$, we define a sub-problem class $\mathcal{F}((x_1, y_1), (b_1, b_2))$ with respect to $\mathcal{F}$ as follows: $f \in \mathcal{F}((x_1, y_1), (b_1, b_2))$ if and only if there exists a $g \in \mathcal{F}$ such that

(1)  $g(x_1, y) = b_1(y)$ for all $y \in \mathcal{Y}$;

(2)  $g(x, y_1) = b_2(x)$ for all $x \in \mathcal{X}$;

(3)  $f$ is a restriction of $g$ on $(\mathcal{X} \setminus \{x_1\}) \times (\mathcal{Y} \setminus \{y_1\})$.

Next, before we show that a sub-problem class $\mathcal{F}((x_1, y_1), (b_1, b_2))$ is c.u.p., we need a lemma to guarantee that after any permutation, the permuted payoff function still contains the same unique Nash Equilibrium[2]. We provide a corollary to Lemma 4. We defer all the proofs to the appendix due to the page limit.

**Corollary 1.** Let $|\mathcal{X}| = |\mathcal{Y}|$ and $\mathcal{F} \subseteq \{g : \mathcal{X} \times \mathcal{Y} \to \mathbb{R}\}$ be any set of two-player zero-sum games with a unique NE. Assume furthermore that $\mathcal{F}$ is c.u.p.. Then, for any game $g \in \mathcal{F}$, let

---

[2]The further explanation for sub-problem classes can be found in Section E.

$(x^*, y^*) \in \mathcal{X} \times \mathcal{Y}$ denote the unique Nash Equilibrium of $g$. For any permutations over $\mathcal{X}, \mathcal{Y}$ denoted by $\sigma \in \mathfrak{S}(\mathcal{X}), \tau \in \mathfrak{S}(\mathcal{Y})$, $(\sigma \otimes \tau) \circ g \in \mathcal{F}$ exhibits the same unique Nash Equilibrium $(x^*, y^*)$. Moreover, if for any $x_0 \neq x^*$ and $y_0 \neq y^*$, the restriction of $g$ on $(\mathcal{X} \setminus \{x_0\}) \times (\mathcal{Y} \setminus \{y_0\})$, denoted by $g|_{(\mathcal{X} \setminus \{x_0\}) \times (\mathcal{Y} \setminus \{y_0\})}$, exhibits the same unique Nash Equilibrium $(x^*, y^*)$.

This permutation essentially swaps the rows and columns in the payoff matrix. Corollary 1 shows that the permutation of rows and columns does not change the optimum (Nash Equilibrium in any given payoff function). Moreover, if we remove the row and column in the given payoff function, which do not contain the unique Nash Equilibrium, then the unique Nash Equilibrium remains the same in the restricted sub-problem.

Next, we prove the sub-problem class is closed under permutation.

**Lemma 1.** If $\mathcal{F}$ is c.u.p., then $\mathcal{F}((x_1, y_1), (b_1, b_2))$ is also c.u.p..

Then, we want to know if we choose different $(x, y)$ in the sub-problem class, are they still isomorphic (i.e., essentially the same problem for any black-box optimisation algorithm)?

**Lemma 2.** For all $(x_1, y_1), (x_2, y_2) \in \mathcal{X} \times \mathcal{Y}$ and $b_1 : \mathcal{Y} \to \mathbb{R}$, $b_2 : \mathcal{X} \to \mathbb{R}$, we have the isomorphism[3]:

$$\mathcal{F}((x_1, y_1), (b_1, b_2)) \simeq \mathcal{F}((x_2, y_2), (b_1, b_2)).$$

Given $(x_t, y_t)$ is the search point by Algorithm $H$ on payoff function $g$ at iteration $t \in \mathbb{N}$ and $(x^*, y^*)$ is the unique NE in a two-player zero-sum game defined by $g : \mathcal{X} \times \mathcal{Y} \to \mathbb{R}$, Theorem 3.1 considers the following query complexity: assume the cost $C_t$ is the unique queries made by Algorithm $H$,

$$T_{\text{LB}}(H, g) := \inf\{C_t > 0 \mid x_t = x^* \text{ or } y_t = y^*\}.$$

Now, we prove our main theorem.

**Theorem 3.1.** *Given $\mathcal{F}$ as a subset of all the payoff functions $g : \mathcal{X} \times \mathcal{Y} \to O$ with a unique Nash Equilibrium where $O \subseteq \mathbb{R}$ and $|\mathcal{X}| = |\mathcal{Y}|$. Let $H$ be an arbitrary (randomised or deterministic) black-box adversarial optimisation algorithm for any $g \in \mathcal{F}$ where $\mathcal{F}$ is closed under permutations. Let $r(H, \mathcal{F})$ be the average (under the uniform distribution on $\mathcal{F}$) of the expected query complexity of $H$ on $g \in \mathcal{F}$ (i.e. $\mathrm{E}(T_{LB}(H, g))$). Then $r(H, \mathcal{F}) = r(H', \mathcal{F})$ for all algorithms $H, H'$.*

*Proof Sketch.* The proof of the main theorem is deferred to the appendix. Briefly, we use a proof by induction on the size of the search space. During the inductive hypothesis, problem class is reduced from $\mathcal{F}$ to $\mathcal{F}((x_1, y_1), (b_1, b_2))$, decreasing the search space from size $N \times N$ to size $(N-1) \times (N-1)$, where $|\mathcal{X}| := N$. Then using Lemma 2 and inductive step, it follows that $r(H, \mathcal{F}) = r(H', \mathcal{F})$ for any two deterministic algorithms $H, H'$, and the claim for randomised algorithms quickly follows from the fact that any randomised algorithm can be viewed as a probability distribution among all deterministic algorithms. $\square$

Theorem 3.1 reveals an important underlying result: All black-box adversarial algorithms can exhibit the same average runtime $r(H, \mathcal{F})$ of all possible problem instances (or problem instances c.u.p.) with a unique Nash Equilibrium in a two-player zero-sum game setting. It is a reasonable result since Theorem 3.1 tells us that if a class of payoff functions with a unique Nash Equilibrium does not change by any permutation on the input space, there is no structure provided for any search heuristic or any optimisation algorithm to use and it cannot help to find the Nash Equilibrium. This is also the reason why the original No Free Lunch theorem for traditional optimisation holds [45, 44]. As concluded by Ho and Pepyne [23], "if anything is possible and occurs with the same probability, then nothing can be expected.

This result is also surprising since Wolpert and Macready [46] and Service and Tauritz [39] both show there exists free lunch with respect to the two solution concepts in Definition 9 and Definition 10. However, our result does contradict the previous result. We consider the performance measure $T(H, g)$ and a different query model considered by the previous work. In particular, Definition 9 and Definition 10 only take the player $x$ into account, while the opponent optimum and different query model and performance measures can make a difference, resulting in either FL or NFL results. In summary, our new NFL theorem highlights the significance of solution concepts and also reveals that adversarial optimisation can exhibit "no-free-lunch", in particular for NE solution concept.

---

[3] Here we define isomorphism following [11] rather than defining it as a usual group or ring isomorphism since we do not require any group or ring structure of $\mathcal{F}$ in the proof.

# 4 Black-Box Complexity of Black-Box Adversarial Optimisation

As shown in the previous NFL theorem, no better universal algorithms exist on structure-free problems (i.e., only assume the payoff function consists of the unique NE). In order for an algorithm to guarantee good performance, it is necessary to restrict the algorithm to classes of games that possess some structure. To compute the Nash equilibrium of certain classes of problems, including Nash equilibrium in a black-box setting, there are many works on analysing the computational complexity for black-box algorithms, and researchers aim to minimise the query complexity and provide more efficient algorithms to compute Nash equilibrium in two-player zero-sum game settings [27, 26, 3, 21, 22] The following questions remain under-explored: How does the performance measure of an algorithm (like query complexity) depend on the size of the search space, and is there any limitation that these algorithms will reach regardless of the problem $g$? We answer these questions here using black-box complexity.

## 4.1 The Unrestricted Black-Box Model and the Black-Box Complexity

This section focuses on adversarial black-box optimisation and study the query complexity of learning Nash equilibrium in a two-player zero-sum-game setting. We refer to [13, 8, 10] as a more detailed introduction to the black-box complexity theory on traditional black-box optimisation. To prove a lower bound that holds for all algorithms, it is necessary first formally to define what constitutes an algorithm. Next, we construct an unrestricted black-box model of adversarial optimisation.

---

**Algorithm 1** An Unrestricted Black-Box Model with Unique Query History

---

**Require:** Search spaces $\mathcal{X}, \mathcal{Y}$.
**Require:** Payoff functions $g : \mathcal{X} \times \mathcal{Y} \to \mathbb{R}$, $h : \mathcal{X} \times \mathcal{Y} \to \mathbb{R}$;
 1: Initialise $(x_0, y_0)$ based on $P_\emptyset$; Initialise $H_0 = \emptyset$ and $C_0 = 0$.
 2: **for** $t = 1, 2, \cdots$ until the termination criterion met **do**
 3:     Choose some probability distribution $P_{I(t)}$, depending only on $I(t)$ where
     $$I(t) := \Pi_{j=1}^{t-1}(x_j, y_j, g(x_j, y_j), h(x_j, y_j)) \in (\mathcal{X} \times \mathcal{Y} \times \mathbb{R} \times \mathbb{R})^{t-1}$$
 4:     Produce a random search point $(x_t, y_t)$ based on $P_{I(t)}$.
 5:     Query the payoffs $g(x_t, y_t), h(x_t, y_t)$
 6:     **if** $(x_t, y_t) \notin H_{t-1}$ **then** $C_t = C_{t-1} + 1$; $H_t = H_{t-1} \cup \{(x_t, y_t)\}$.
 7:     **else** $C_t = C_{t-1}$; $H_t = H_{t-1}$.

---

Algorithm 1 defines a class of algorithms subject to various probability distributions and samples the new strategy pair based on previous pairs and their payoffs. The initial search point $(x_0, y_0)$ is independent of the problem, so we can choose any probability distribution $P_\emptyset$ to initialise the algorithm. Subsequent strategy pairs are obtained by asking the oracle to apply a given variation operator to all previously queried search points subject to sample probability distribution $P_{I(t)}$. By specifying different sample probability distribution $P_{I(t)}$, Algorithm 1 represents various black-box optimisation algorithms, including several adversarial search (also called competitive coevolutionary) algorithms [46, 32] and randomised algorithms FINDPSNE (designed to learn the NE in bimatrix games) [27]. Note that the model only considers the cost of unique queries made by the algorithm (i.e. we check the search point in the previous query history $H_t$ in line 6). We assume that an algorithm which makes the same query twice is only charged for the cost of one of the queries.

Now, we define the query complexity (or runtime) of black-box adversarial optimisation algorithms by extending the idea of the traditional single-objective black-box optimisation algorithm in [13, 43] and assuming $h = -g$ (i.e. zero-sum game) in this case.

**Definition 4.** Given any unrestricted black-box algorithm with unique query history $A$ and the payoff function $g : \mathcal{X} \times \mathcal{Y} \to \mathbb{R}$, $T(A, g)$ is the query complexity of $A$ with respect to $g$ and $(x_t, y_t)$ is the search point generated by $A$ if

$$T(A, g) := \inf\{dC_t \in \mathbb{N} \mid (x_t, y_t) \in arg \max_{x \in \mathcal{X}} \min_{y \in \mathcal{Y}} g(x, y)\}$$

where $d \in \{1, 2\}$. If the game is zero-sum, then $d = 1$. Otherwise, $d = 2$

Note that $T(A, g) \in \mathbb{R} \cup \{\infty\}$ is the number of payoff evaluations until $A$ queries for the first time some $(x^*, y^*) \in \arg\max_{x \in \mathcal{X}} \min_{y \in \mathcal{Y}} g(x, y)$. Now, we can define what black-box complexity means with respect to a given class of adversarial optimisation algorithms and problem classes.

**Definition 5.** For a class $\mathcal{G}$ of payoff functions $g : \mathcal{X} \times \mathcal{Y} \to \mathbb{R}$, the $A$-black-box complexity of $\mathcal{G}$ is defined as $T(A, \mathcal{G}) := \sup_{g \in \mathcal{G}} T(A, g)$, the runtime of $A$ under the worst-case scenario on $\mathcal{G}$. Then, the $\mathcal{A}$-black-box complexity of $\mathcal{G}$ is $T(\mathcal{A}, \mathcal{G}) := \inf_{A \in \mathcal{A}} T(A, \mathcal{G}) = \inf_{A \in \mathcal{A}} \sup_{g \in \mathcal{G}} T(A, g)$; the best or minimum complexity among all $A \in \mathcal{A}$ with respect to $\mathcal{G}$. If $\mathcal{A}$ is the whole class of all black-box algorithms, we denote $T(\mathcal{A}, \mathcal{G})$ the unrestricted black-box complexity of $\mathcal{G}$.

We want to point out the difference between these two definitions. Definition 4 considers the query complexity for a particular algorithm and problem instance, and Definition 5 considers the black-box complexity for the best possible query complexity of all possible given algorithms under the worst-case scenario. The traditional black-box complexity theory characterises the difficulty of a certain class of problems and explores the limitations of the given black-box optimisation algorithms. We expect our extension to adversarial optimisation can provide similar insights as well.

## 4.2 A General Lower Bound for Black-Box Adversarial Optimisation

We first provide a lower bound of query complexity for a general class of black-box adversarial optimisation problems.

**Theorem 4.1.** *Let $\mathcal{X}$ and $\mathcal{Y}$ be any finite sets. Assume that $B \subset \mathbb{R}$ with $k := |B| \geq 2$. Consider any class of two-player zero-sum games $\mathcal{G} \subset \{g : \mathcal{X} \times \mathcal{Y} \to B\}$ such that for all $(x, y) \in \mathcal{X} \times \mathcal{Y}$, there exists a game $g_{x,y} \in \mathcal{G}$ which has $(x, y)$ as unique, pure Nash Equilibrium. Then, the class $\mathcal{G}$ has black box complexity at least $\lceil \log_k |\mathcal{X} \times \mathcal{Y}| \rceil - 1$.*

## 4.3 A General Lower Bound for Two-player Zero-Sum Bimatrix Games

In this subsection, we employ Yao's Principle and No Free Lunch Theorem to provide a general lower bound for query complexity of searching Nash Equilibrium in zero-sum bimatrix games, and this leads to a sharper lower bound compared to the previous result in [27].

**Theorem 4.2.** *Let $\mathcal{A}$ be the set of all randomised algorithms defined by Algorithm 1 and $T(\mathcal{A}, P)$[4] denote the query complexity of $\mathcal{A}$ with respect to the input payoff matrix $P$ for a two-player zero-sum game. Then, there exists an input matrix $P \in \mathbb{R}^{n \times n}$ with a unique pure Nash equilibrium $(x^*, y^*)$ such that $\mathrm{E}(T(\mathcal{A}, P)) \geq (n + 1)/2$. Thus, the black-box complexity with respect to $\mathcal{A}$ of the problem class consisting of all bimatrix games with a unique Nash Equilibrium is at least $(n + 1)/2$.*

Theorem 4.2 provides a sharper lower bound by a multiplicative factor 4 compared with the current best bound by [27]. This result also demonstrates that in a two-player zero-sum bimatrix game (with an $n \times n$ payoff matrix), there are complex instances where no randomised algorithm can achieve better than $O(n)$ query complexity unless additional problem structure is provided.

## 4.4 Applications on Two-Player Zero-Sum Games

### 4.4.1 Introduction to Binary Voting Games

This section provides some example applications of our black-box complexity results. Voting games are popular games studied in game theory, computational social choice theory [5, 4, 16] and Boolean games [18]. Voting is considered a fundamental tool for analysing multi-agent systems [16, 1]. We start with simple binary voting games in which the outcome or payoff is 0 or 1 (or $-1$ and 1). These games also play a role in the analysis of Boolean functions [31].

Convergence to NE in plurality voting has been studied from the perspective of social choice theory and researchers specify certain conditions to guarantee the voting games to converge to NEs [28]. Some natural question arises: are there any randomised algorithms that can find these NEs in voting games efficiently? What are the limitations of these black-box optimisation algorithms? Using the black-box complexity analysis, we can answer the questions about the efficiency/inefficiency of black-box optimisation algorithms on binary voting games.

---

[4]Note that $T(A, P) = T(A, g)$ where $g(x, y) := e_y^T P e_x$ with $e_x, e_y$ denote the elementary probability distribution over probability simplex $\Delta_{\{0,1\}^n}$

We formulate binary voting games in the context of adversarial optimisation as follows. Consider two parties represented by vectors $x, y \in \{0, 1\}^n$ where $n \in \mathbb{N}$. Each group has $n$ members that either "in favour" (encoded by 1) or "against" (encoded by 0) a particular proposal or decision. One group seeks a strategy $x^*$ that maximises its minimum gains against any strategy of the other group, while the other group seeks a strategy $y^*$ such that its choice minimises the maximum gains of the first group. It essentially forms a two-player zero-sum game.

**Definition 6.** For $\mathcal{X} = \mathcal{Y} = \{0, 1\}^n$, the payoff function DIAGONAL $: \mathcal{X} \times \mathcal{Y} \to \{-1, 1\}$ is

$$\text{DIAGONAL}(x, y) := \begin{cases} 1 & |y|_1 \le |x|_1 \\ -1 & \text{otherwise.} \end{cases}$$

In Definition 6, we present the votes of both groups by binary bitstrings and the payoff $g$ can be viewed as a binary voting game where the payoff only depends on which group has the stronger majority "influence". If one group has a stronger 'in favour' "influence" in the sense of the number of the support votes (i.e. the number of 1 in the encoding binary bitstring), then we get a payoff 1 and $-1$ otherwise. We are interested in computing NE in these two-player zero-sum games, i.e. solving $(x^*, y^*) \in \arg\max_{x \in \mathcal{X}} \min_{y \in \mathcal{Y}} g(x, y)$. Notice that in DIAGONAL, $(x_n, y_n) = (1^n, 1^n)$ is the unique NE optimum. In this optimum, neither of the two groups is willing to deviate from affecting their payoff $g(x_n, y_n)$ anymore. This exactly coincides with the definition of NE.

Next, we consider a different binary voting game, denoted by PLATEAU. To make the binary voting game more challenging, we introduce some plateaus in games.

**Definition 7.** For $\mathcal{X} = \mathcal{Y} = \{0, 1\}^n$, a constant $\delta \in (0, 1)$, the payoff function PLATEAU $: \mathcal{X} \times \mathcal{Y} \to \{-1, 1\}$ is defined as

$$\text{PLATEAU}(x, y) := \begin{cases} f(y) & \text{if } ||x|_1 - \frac{n}{2}| < \frac{\delta n}{2} \\ g(x, y) & \text{otherwise} \end{cases}$$

where $f : \mathcal{Y} \to \{-1, 1\}$ and $g : \mathcal{X} \times \mathcal{Y} \to \{-1, 1\}$ are any functions such that the NE of PLATEAU is $(x^*, y^*) \notin \{(x, y) \mid ||x|_1 - \frac{n}{2}| < \frac{\delta n}{2}\}$.

Definition 7 introduces a plateau when comparing the "influence" between two groups and defines a general class of pseudo-Boolean benchmarks with a plateau. Imagine a committee deciding on a new policy where there are two groups with equal voting power. If the votes from group $\mathcal{X}$ are balanced or nearly balanced (within the plateau), then the votes from the second group ($\mathcal{Y}$) come into play. It is like their votes are the tiebreaker. If the second group votes in favour, then the policy passes; if they vote against it, then it fails. If the votes from group $\mathcal{X}$ are outside the plateau, then the payoff is not restricted to be determined by $y \in \mathcal{Y}$.

Finally, we define game instances generated by $(u, v)$ where $u, v \in \{0, 1\}^n$.

**Definition 8** $((u, v)$-game instance$)$**.** For all $u, v \in \{0, 1\}^n$, given $f : \{0, 1\}^n \times \{0, 1\}^n \to \mathbb{R}$, we define the $(u, v)$-instance of $f$, denoted by $f_{(u,v)}$, as $f_{(u,v)}(x, y) := f(u \oplus x, v \oplus v)$.

We can see that for any $u, v \in \{0, 1\}^n$, $f_{(u,v)}$ generates the same payoff landscape as $f$. In this paper, $f$ will be either DIAGONAL or PLATEAU. We defer more details to Section G[5].

### 4.4.2 Black-Box Complexity of Learning Nash Equilibrium in Binary Voting Games

First, let us illustrate how Theorem 4.1 and Theorem 4.2 work on these simple examples. If we consider a general class of black-box optimisation algorithms defined by Algorithm 1 on binary voting games with a unique Nash Equilibrium, then we provide a general lower bound of black-box complexity as follows.

**Theorem 4.3.** *The black-box complexity with respect to Algorithm 1 of the binary voting games with problem size $n \in \mathbb{N}$ and a unique Nash Equilibrium is $e^{\Omega(n)}$.*

Theorem 4.3 means that there exist no universal good black-box optimisation algorithms defined by Algorithm 1 that can solve all binary voting games with unique Nash Equilibrium efficiently, (i.e. with polynomial query complexity of the problem size). To yield a better performance of black-box

---

[5]We defer the definition of xor $\oplus$ to Section G in the supplementary material.

optimisation algorithms on binary voting games, we need to specify the problem class we work on. Next, we consider DIAGONAL and explore the black-box complexity with respect to the class of black-box optimisation algorithms $\mathcal{A}$ defined by Algorithm 1 of DIAGONAL. This reveals that DIAGONAL is a feasible benchmark problem for black-box adversarial optimisation algorithms.

**Theorem 4.4.** *Given the game class*

$$\text{DIAGONAL}_n := \{\text{DIAGONAL}_{(u,v)} \mid (u,v) \in \{0,1\}^n \times \{0,1\}^n\},$$

*the black-box complexity with respect to Algorithm 1 of* $\text{DIAGONAL}_n$ *is* $\Theta(n)$.

Theorem 4.4 implies that $\text{DIAGONAL}_n$ is a sensible maximin benchmark for testing black-box optimisation algorithms. It means that if we restrict the problem class to a certain class with a specific structure, then it is possible to solve them in polynomial query complexity.

We have seen the black-box complexity results on $\text{DIAGONAL}_n$. Next, we start to consider more challenging binary voting games, PLATEAU. We are interested in whether there is any efficient black-box adversarial optimisation that can solve $\text{PLATEAU}_n$ in polynomial query complexity (i.e. $O(n)$). To answer this question, we need to compute its black-box complexity.

**Theorem 4.5.** *Given the game class,*

$$\text{PLATEAU}_n := \{\text{PLATEAU}_{(u,v)} \mid (u,v) \in \{0,1\}^n \times \{0,1\}^n\},$$

*the black-box complexity with respect to the class of algorithms defined by Algorithm 1 of* $\text{PLATEAU}_n$ *is* $e^{\Omega(n)}$.

Theorem 4.5 implies that all black-box adversarial optimisation algorithms defined by unrestricted model (i.e. Algorithm 1) have exponential runtime on $\text{PLATEAU}_n$. They need at least an exponentially large query complexity with respect to the problem size $n$. It is evident that PLATEAU is too challenging that it may not be a proper benchmark for black-box adversarial optimisation algorithms.

### 4.4.3 Summary

Our study introduces the concept of black-box complexity in binary voting games, providing insights into the challenges faced by general black-box adversarial optimisation algorithms. These examples illustrate two kinds of problems within the general class of binary voting games with unique NE: the polynomial-solvable class (i.e., there exists an algorithm that can solve all problem instances of this class in polynomial runtime) and the non-polynomial-solvable class (i.e., there exists no algorithm that can solve all problem instances of this class in polynomial runtime).

Theorem 4.3 rigorously show that no universal algorithm can efficiently learn the unique Nash Equilibrium (NE) in these games due to their structure-free nature, where efficiency means small query complexity. However, when assuming specific problem structures, such as $\text{DIAGONAL}_n$, it is possible to come up with a better algorithm which achieves better polynomial query complexity as shown in Theorem 4.4. $\text{DIAGONAL}_n$ can be a promising benchmark for evaluating black-box algorithms. Additional assumptions on the payoff function, like those in $\text{PLATEAU}_n$ problems, proved in Theorem 4.5, do not lower the difficulty of the problems. This emphasises the need for a more careful selection of benchmarks. Black-box complexity emerges as a valuable tool for distinguishing between potentially easy and hard problem instances, guiding the design of black-box algorithms.

## 5 Further Related Work

### 5.1 Co-evolutionary Search Heuristics

Next, we provide some practical examples of adversarial optimisation. There are various adversarial search heuristics, including competitive co-evolutionary algorithms (CoEAs) [35, 26]. CoEAs are a class of algorithms applied in game-theoretic and strategic adversarial optimisation scenarios. For example, CoEAs are used to solve maximin optimisation in a cybersecurity context [32] and to enhance GANs by using a co-evolutionary approach for image translation [41]. Similarly, the neural architecture search system Lipizzaner employs co-evolutionary adversarial search to find suitable neural architectures for GANs [42].

Although several applications of adversarial (or co-evolutionary) search exist, there is limited theoretical literature on this topic. Lehre [26] demonstrated that the running time of the co-evolutionary

algorithm PDCoEA on instances of the discrete bilinear problem is polynomial. Later, Hevia Fajardo et al. [22] showed a weakness of RLS-PD and the sufficiency of a simple archive to prevent evolutionary forgetting. On the other hand, regarding the general black-box optimisation framework, Wolpert and Macready [46] showed there exists a "free lunch" in such adversarial (or co-evolutionary) optimisation setting – there exist some algorithms have better performance than others averaged across all possible problem instances in adversarial optimisation with respect to the maximin solution concept (see Definition 10).

## 5.2 Query Complexity of Learning in Games

The query complexity of various solution concepts in zero-sum and general-sum multi-player games has been well studied (for an overview, see [2, 27]). It is well-known that converging to an exact (mixed strategy) Nash Equilibrium in general-sum matrix games is PPAD-complete [7]. Chen and Deng [6] provide a simplified proof of the computational complexity of Nash Equilibrium in two-player games. Significant focus has also been placed on how the game dynamics converge to the Nash equilibrium or approximate it [27, 25, 17, 34].

Panageas et al. [34] considered the complexity of fictitious play in two-player potential games with a unique Nash Equilibrium (NE). They proved the existence of a hard instance which requires query complexity $\Omega\left(4^n\left((n/2-2)!\right)^4\right)$ where $n$ refers to the number of actions. They showed that fictitious play can take exponential time (in the number of strategies) to reach a unique Nash Equilibrium, even when the game is restricted to two agents and arbitrary tie-breaking rules, by constructing a two-player coordination game with a unique Nash Equilibrium. Unlike previous work, our paper focuses on a more general class of games: the entire class of two-player zero-sum games with a unique NE rather than just potential games. We show that all black-box adversarial algorithms, including fictitious play, exhibit the same average performance across all problem instances in terms of query complexity.

# 6 Conclusion and Discussion

Utilising the tools from game theory and Yao's principle, we rigorously prove the impossibility results for a universally effective adversarial algorithm applicable across various problem classes in black-box adversarial optimisation. We emphasise the impact of solution concept selection on the feasibility of a "free lunch" in adversarial optimisation. Additionally, we introduce the notion of black-box complexity in black-box adversarial optimisation and characterise the difficulty of learning the unique optimum in adversarial optimisation and solving two-player zero-sum games.

The results from this paper build up a foundation for future studies on the strengths and limitations of adversarial optimisation. More specifically, it highlights the need for more comprehensive benchmarks and careful selections of solution concepts when using any black-box adversarial optimisation algorithms. Moreover, no black-box optimisation algorithm for learning the Nash equilibrium in two-player zero-sum games can exceed the logarithmic complexity relative to search space size. Meanwhile, no algorithm can solve any bimatrix game with unique NE faster than the linear query complexity in terms of the size of input payoff matrices.

Although our work makes a first step towards the new NFL and BBC results on black-box adversarial optimisation, we want to point out some limitations of our current work and list them as our future work. Firstly, our theoretical results build on discrete exponential large search spaces rather than countably infinite (e.g. $\mathbb{N}$) or uncountable infinite (e.g. $\mathbb{R}$) sets. To generalise our results to infinite sets, we might require further assumptions on our search spaces. Secondly, our NFL focuses on two-player zero-sum games with unique NE.

Future direction of our work includes extending Theorem 3.1 to other solution concepts like mixed strategy Nash Equilibrium or exploring different possible solution concepts which may exhibit free lunch or not. Additionally, it is interesting to generalise NFL and BBC analysis to zero-sum games with multiple NEs and more general search spaces. Finally, it is interesting to analyse other black-box models, such as unbiased black-box complexity models, to characterise the difficulty of adversarial problems that different classes of search heuristics can solve.

## Acknowledgements

We would like to thank Dr Alistair Benford and Dr Mario Alejandro Hevia Fajardo for the fruitful discussion and comments on an earlier draft of this paper. This work was supported by a Turing AI Fellowship (EPSRC grant ref EP/V025562/1).

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

## Appendix / Supplementary material

## Contents

## A  Two Other Solution Concepts

**Definition 9** (Maximisation Over All Test Cases [39] )**.**  Suppose a set of candidate solutions, $\mathcal{X}$, and a set of test cases, $\mathcal{Y}$. Given an interaction function or payoff function, $g : \mathcal{X} \times \mathcal{Y} \to O$, where $O$ is an ordered set which determines the outcome of candidate solution $\mathcal{X}$ on test $\mathcal{Y}$, the solution concept is defined by: $S = \{x^* \in \mathcal{X} \mid \forall x \in \mathcal{X}, \forall y \in \mathcal{Y}, g(x^*, y) \geq g(x, y)\}$.

Next, we define the Maximin solutions are those solution configurations which maximise the minimum outcome over all test cases.

**Definition 10** (Maximin [46])**.**  Suppose a set of candidate solutions, $\mathcal{X}$, and a set of test cases, $\mathcal{Y}$. Given an interaction function or payoff function, $g : \mathcal{X} \times \mathcal{Y} \to O$, where $O$ is an ordered set which determines the outcome of candidate solution $\mathcal{X}$ on test $\mathcal{Y}$, the solution concept is defined by:

$$S = \{x^* \in \mathcal{X} \mid \forall x \in \mathcal{X}, \min_{y \in \mathcal{Y}} g(x^*, y) \geq \min_{y \in \mathcal{Y}} g(x, y)\}.$$

Notice that in Definition 9, this solution concept considers the candidate solution $x$ with respect to all test cases $y$. Definition 10 considers the candidate solution with respect to the worst-case scenario. If $\mathcal{Y}$ is compact, then these two solution concepts are essentially the same. Compared with Nash Equilibrium in Definition 1, these two solution concepts only focus on finding the best candidate solution and the algorithm is not required also to produce an opposing "optimal" solution $y^*$.

## B  Technical Lemmas for Games with Unique Nash Equilibrium

We adopt the notation in [27].

**Definition 11.**  Given a two-player zero-sum game on a payoff matrix $P \in \mathbb{R}^{n \times n}$. Let $\Delta_n$ denote the $n$-dimensional probability simplex (i.e. for $p \in \Delta_n$, $\sum_{i=1}^{n} p_i = 1$ and $p_i \geq 0$). A pair $(p^*, q^*)$ is a (mixed strategy) Nash Equilibrium of an input matrix $P \in \mathbb{R}^{n \times n}$ if and only if the following holds for all $(p, q) \in \Delta_n \times \Delta_n$:

$$\langle p, Pq^* \rangle \leq \langle p^*, Pq^* \rangle \leq \langle p^*, Pq \rangle.$$

In this paper, we are only interested in Pure Strategy Nash Equilibrium.

**Definition 12.**  For any payoff matrix $P \in \mathbb{R}^{n \times n}$, let $V_P^\star := \max_{p \in \Delta_n} \min_{q \in \Delta_n} \langle p, Pq \rangle$. In particular, we say an entry $(i^*, j^*) \in [n] \times [n]$ is a unique pure strategy Nash Equilibrium if $(e_{i^*}, e_{j^*}) \in \Delta_n \times \Delta_n$ is a unique Nash Equilibrium of $P$ where $e_{i^*}, e_{j^*}$ are the unit probability vector.

We refer to the following lemma from [40].

**Lemma 3** (Lemma 1 in [27])**.**  Consider a matrix $A \in \mathbb{R}^{n \times n}$ with a unique Nash equilibrium $(p^\star, q^\star)$. The following conditions hold:

1. The support sizes of $p^\star$ and $q^\star$ are equal (i.e., $|\text{supp}(p^\star)| = |\text{supp}(q^\star)|$).

2. $V_A^\star = \langle e_i, Aq^\star \rangle$ for all $i \in \text{supp}(p^\star)$, and $V_A^\star > \langle e_i, Aq^\star \rangle$ for all $i \notin \text{supp}(p^\star)$.

3. $V_A^\star = \langle p^\star, Ae_j \rangle$ for all $j \in \text{supp}(q^\star)$, and $V_A^\star < \langle p^\star, Ae_j \rangle$ for all $j \notin \text{supp}(q^\star)$.

Lemma 3 allows us to derive the following result, which is proved in Appendix B.1 in [27].

**Lemma 4** (Lemma 2 in [27])**.**  Consider a matrix $P \in \mathbb{R}^{n \times n}$ with a unique Nash equilibrium $(p^\star, q^\star)$. Consider a submatrix $M$ of $P$ with row index set $I_X$ and column index set $I_Y$. If $\text{SUPP}(p^\star) \subseteq I_X$ and $\text{SUPP}(q^\star) \subseteq I_Y$, then $(\hat{p}, \hat{q})$ is the unique Nash equilibrium of $M$ where $(\hat{p})_i = (p^\star)_i$ for all $i \in I_X$ and $(\hat{q})_j = (q^\star)_j$ for all $j \in I_Y$. Moreover, $V_M^\star = V_P^\star$.

# C   Analytic Tools from Probability Theory and Analysis of Randomised Algorithms

## C.1   Yao's Principle

In this section, we provide the mathematical tool that we use in the rest of the analysis. Yao's Principle [47] can be used to give lower bounds of a class of randomised algorithms.

**Theorem C.1** (Yao's Principle [47]). *Let $\Pi$ be a problem with a finite set $\mathcal{I}$ of input instances (of a fixed size) permitting a finite set $\mathcal{A}$ of deterministic algorithms. Let $p$ be a probability distribution over $\mathcal{I}$ and $q$ be a probability distribution over $\mathcal{A}$. Then,*

$$\min_{A \in \mathcal{A}} \mathrm{E}(T(I_p, A)) \leq \max_{I \in \mathcal{I}} \mathrm{E}(T(I, A_q))$$

*where $I_p$ denotes a random input chosen from $\mathcal{I}$ according to $p$, $A_q$ a random algorithm chosen from $\mathcal{A}$ according to $q$ and $T(I, A)$ denote the runtime of algorithm $A$ on input $I$.*

## C.2   Markov's inequality and Chernoff's bound

**Theorem C.2** (Markov's inequality). *Given a non-negative random variable $X$ and $a > 0$, $\Pr(X \geq a) \leq \frac{\mathrm{E}(X)}{a}$.*

**Theorem C.3** (Chernoff's bound). *Given $X := \sum_{i=1}^{n} X_i$ where $X_i \sim Ber(p_i)$ where $Ber(p_i)$ is a Bernoulli random variable with probability $p_i$ and all $X_i$ are independent. Let $\mu := \mathrm{E}(X) = \sum_{i=1}^{n} p_i$. Then, for $\delta \in (0, 1)$,*

$$\Pr(|X - \mu| \geq \delta\mu) \leq 2e^{-\mu\delta^2/3}.$$

# D   Omitted Proofs

**Corollary 2.** Let $|\mathcal{X}| = |\mathcal{Y}|$ and $\mathcal{F} \subseteq \{g : \mathcal{X} \times \mathcal{Y} \to \mathbb{R}\}$ be any set of two-player zero-sum games with a unique NE. Assume furthermore that $\mathcal{F}$ is c.u.p.. Then, for any game $g \in \mathcal{F}$, let $(x^*, y^*) \in \mathcal{X} \times \mathcal{Y}$ denote the unique Nash Equilibrium of $g$. For any permutations over $\mathcal{X}, \mathcal{Y}$ denoted by $\sigma \in \mathfrak{S}(\mathcal{X}), \tau \in \mathfrak{S}(\mathcal{Y})$, $(\sigma \otimes \tau) \circ g \in \mathcal{F}$ exhibits the same unique Nash Equilibrium $(x^*, y^*)$. Moreover, if for any $x_0 \neq x^*$ and $y_0 \neq y^*$, the restriction of $g$ on $(\mathcal{X} \setminus \{x_0\}) \times (\mathcal{Y} \setminus \{y_0\})$, denoted by $g|_{(\mathcal{X} \setminus \{x_0\}) \times (\mathcal{Y} \setminus \{y_0\})}$, exhibits the same unique Nash Equilibrium $(x^*, y^*)$.

*Proof of Lemma 1.* We employ the result in [27] (See Lemma 4). For any permutations on $\mathcal{X}, \mathcal{Y}$ denoted by $\sigma \in \mathfrak{S}(\mathcal{X}), \tau \in \mathfrak{S}(\mathcal{Y})$, $(\sigma \otimes \tau) \circ g \in \mathcal{F}$ is defined as for any $x \in X, y \in \mathcal{Y}$,

$$(\sigma \otimes \tau) \circ g(x, y) = g(\sigma(x), \tau(y)).$$

We consider the payoff matrix defined by $P = (p_{i,j})$ where $p_{i,j} = g(x_i, y_j)$ with $x_i \in \mathcal{X}, y_j \in \mathcal{Y}$. We denote row index set by $I_{\mathcal{X}} \subseteq \mathbb{N}$ and column index set by $I_{\mathcal{Y}} \subseteq \mathbb{N}$. Notice that $|I_{\mathcal{X}}| = |\mathcal{X}| = |\mathcal{Y}| = |I_{\mathcal{Y}}|$. We denote the row index by $i \in I_{\mathcal{X}}$ and the column index by $j \in I_{\mathcal{Y}}$. $P \in \mathbb{R}^{|I_{\mathcal{X}}| \times |I_{\mathcal{Y}}|}$ is the payoff matrix of two-player zero-sum game $g$ with a unique Nash Equilibrium $(x^*, y^*)$. Let us denote the row index of $x^*$ in the payoff matrix $P$ by $i_*$ and the column index of $y^*$ by $j_*$ in the payoff matrix $P$. So $(e_{i^*}, e_{j^*}) \in \Delta_n \times \Delta_n$ is the unique Nash Equilibrium in the form of probability vector.

Now, given any permutation $\sigma, \tau$, we consider a submatrix $P'$ such that $P' = (p_{\sigma(i), \tau(j)})$ with a new row index set $I'_{\mathcal{X}} \subseteq \mathbb{N}$ and column index set $I'_{\mathcal{Y}} \subseteq \mathbb{N}$. Since $\sigma$ only shuffles around the row indices in $I_{\mathcal{X}}, I'_{\mathcal{X}} \simeq I_{\mathcal{X}}$ with the isomorphism $\sigma$. Similarly, we have $I'_{\mathcal{Y}} \simeq I_{\mathcal{Y}}$ with the isomorphism $\tau$. Now, we can see that $i^* = \mathrm{SUPP}(e_{i^*}) \in I'_{\mathcal{X}}$ and $j^* = \mathrm{SUPP}(e_{j^*}) \in I'_{\mathcal{Y}}$, so $(\hat{p}, \hat{q}) \in \Delta_n \times \Delta_n$ is the unique Nash Equilibrium of $P'$ (in the form of a probability vector) where $(\hat{p})_i = (e_{i^*})_i$ for all $i \in I'_{\mathcal{X}}$ and $(\hat{q})_j = (e_{j^*})_j$ for all $j \in I'_{\mathcal{Y}}$. In other words, $(\hat{p}, \hat{q}) = (e_{i^*}, e_{j^*})$. So by using Lemma 4, we can conclude that $(x^*, y^*) \in \mathcal{X} \times \mathcal{Y}$ is the unique Nash Equilibrium of $(\sigma \otimes \tau) \circ g$.

For the second claim, we construct a submatrix $Q$ such that $Q = (q_{i,j})$ where $q_{i,j} = g(x_i, y_j)$ with $x_i \in \mathcal{X} \setminus \{x_0\}, y_j \in \mathcal{Y} \setminus \{y_0\}$. We denote the row index set $I_{\mathcal{X} \setminus \{x_0\}}$ and the column index set $I_{\mathcal{Y} \setminus \{y_0\}}$. Since $x_0 \neq x_*$ and $y_0 \neq y_*$, $i^* = \mathrm{SUPP}(e_{i^*}) \in I_{\mathcal{X} \setminus \{x_0\}}$ and $j^* = \mathrm{SUPP}(e_{j^*}) \in I_{\mathcal{Y} \setminus \{y_0\}}$, by using

Lemma 4 again, we can conclude that $(x^*, y^*)$ is the unique Nash Equilibrium of $g|_{(\mathcal{X} \setminus \{x_0\}) \times (\mathcal{Y} \setminus \{y_0\})}$. We complete the proof. $\square$

**Lemma 1.** If $\mathcal{F}$ is c.u.p., then $\mathcal{F}((x_1, y_1), (b_1, b_2))$ is also c.u.p..

*Proof of Lemma 1.* For any $\sigma' \in \mathfrak{S}(\mathcal{X} \setminus \{x_1\}), \tau' \in \mathfrak{S}(\mathcal{Y} \setminus \{y_1\})$ and any $g' \in \mathcal{F}((x_1, y_1), (b_1, b_2))$, we want to show $(\sigma'\tau')g' \in \mathcal{F}((x_1, y_1), (b_1, b_2))$. We consider the extensions of each permutation. We define $\sigma : \mathcal{X} \to \mathcal{X}$ by its restriction on $\mathcal{X} \setminus \{x_1\}$ as

$$\sigma|_{\mathcal{X} \setminus \{x_1\}} = \sigma' \text{ and } \sigma(x_1) = x_1.$$

We define $\tau : \mathcal{Y} \to \mathcal{Y}$ by its restriction on $\mathcal{Y} \setminus \{y_1\}$ is

$$\tau|_{\mathcal{X} \setminus \{y_1\}} = \tau' \text{ and } \tau(y_1) = y_1.$$

Now, let $g$ be the extension of $g'$ which satisfies (1), (2) and (3) of Definition 3 (such an extension $g$ exists since we take $g' \in \mathcal{F}((x_1, y_1), (b_1, b_2))$ and it follows from the definition of sub-problem class). As $g \in \mathcal{F}$, so are $(\sigma\tau)g \in \mathcal{F}$. Thus, we construct $(\sigma\tau)g$ as an extension of $(\sigma'\tau')g'$ and $(\sigma\tau)g$ satisfies (1), (2) and (3) of Definition 3. So $(\sigma'\tau')g' \in \mathcal{F}((x_1, y_1), (b_1, b_2))$. $\square$

**Lemma 2.** For all $(x_1, y_1), (x_2, y_2) \in \mathcal{X} \times \mathcal{Y}$ and $b_1 : \mathcal{Y} \to \mathbb{R}$, $b_2 : \mathcal{X} \to \mathbb{R}$, we have the isomorphism[6]:

$$\mathcal{F}((x_1, y_1), (b_1, b_2)) \simeq \mathcal{F}((x_2, y_2), (b_1, b_2)).$$

*Proof of Lemma 2.* To prove the isomorphism, we need to find a bijection between these two sets. We consider the following map between sets with , defined by:

$$\phi : \mathcal{F}((x_1, y_1), (b_1, b_2)) \to \mathcal{F}((x_2, y_2), (b_1, b_2)) \text{ where } \phi(g') = h' \text{ such that}$$

for $(x, y) \in (\mathcal{X} \setminus \{x_1, x_2\}) \times (\mathcal{Y} \setminus \{y_1, y_2\})$,

$$h'(x, y) = \phi(g')(x, y) = g'(x, y).$$

For $y \in \mathcal{Y} \setminus \{y_2\}$, $h'(x_1, y) = \phi(g')(x_2, y) = g'(x_2, y)$.
For $x \in \mathcal{Y} \setminus \{x_2\}$, $h'(x, y_1) = \phi(g')(x, y_2) = g'(x, y_2)$ Let $h$ be the extension of $h'$ defined by

$$h(x_2, y) = b_1(y), h(x, y_2) = b_2(x)$$

For all $(x, y) \in (\mathcal{X} \setminus \{x_2\}) \times (\mathcal{Y} \setminus \{y_2\})$,

$$h(x, y) = h'(x, y)$$

Now, we notice that $h = (\pi_x \pi_y)g$ where $\pi_x, \pi_y$ are two transpositions in $\mathfrak{S}(\mathcal{X}), \mathfrak{S}(\mathcal{Y})$ defined by

$$\pi_x(x_1) = x_2, \pi_x(x_2) = x_1$$
$$\pi_y(y_1) = y_2, \pi_y(y_2) = y_1$$

and for all $x' \in \mathcal{X} \setminus \{x_1, x_2\}$ and for all $y' \in \mathcal{Y} \setminus \{y_1, y_2\}$,

$$\pi_x(x') = x', \pi_y(y') = y'$$

Since $g \in \mathcal{F}$, so is $h = (\pi_x \pi_y)g \in \mathcal{F}$. So $h'$ has the extension $h$ satisfies (1), (2) in Definition 3. Also, $h' = \phi(g')$ is uniquely determined by $g'$ and the argument would also hold when swapping $h', g'$ with the map $\phi^{-1}$. So we prove there exists an isomorphism between these two sub-problem classes. $\square$

**Theorem 3.1.** *Given $\mathcal{F}$ as a subset of all the payoff functions $g : \mathcal{X} \times \mathcal{Y} \to O$ with a unique Nash Equilibrium where $O \subseteq \mathbb{R}$ and $|\mathcal{X}| = |\mathcal{Y}|$. Let $H$ be an arbitrary (randomised or deterministic) black-box adversarial optimisation algorithm for any $g \in \mathcal{F}$ where $\mathcal{F}$ is closed under permutations. Let $r(H, \mathcal{F})$ be the average (under the uniform distribution on $\mathcal{F}$) of the expected query complexity of $H$ on $g \in \mathcal{F}$ (i.e. $\mathrm{E}(T_{LB}(H, g))$). Then $r(H, \mathcal{F}) = r(H', \mathcal{F})$ for all algorithms $H, H'$.*

---

[6]Here we define isomorphism following [11] rather than defining it as a usual group or ring isomorphism since we do not require any group or ring structure of $\mathcal{F}$ in the proof.

*Proof of Theorem 3.1.* Recall that the query complexity of $H$ on $g$ is $T_{\mathrm{LB}}(H, g) := \inf\{t > 0 \mid x_t = x^*$ or $y_t = y^*\}$ where $(x_t, y_t)$ is the search point by Algorithm $H$ on payoff function $g$ and $(x^*, y^*)$ is the unique Nash Equilibrium in a two-player zero-sum game defined by $g : \mathcal{X} \times \mathcal{Y} \to O$. We denote the set of all functions from a set $\mathcal{X}$ to a set $O$ by $\mathcal{H}(\mathcal{X}, O)$ and the set of all well-defined functions from a set $\mathcal{Y}$ to a set $O$ by $\mathcal{H}(\mathcal{Y}, O)$.

Let $\mathcal{F}$ be a class of games with all the payoff functions $g : \mathcal{X} \times \mathcal{Y} \to O$ with a unique NE, and assume that $\mathcal{F}$ is closed under permutation. For all $(x_0, y_0) \in \mathcal{X} \times \mathcal{Y}$, define

$$B_{x_0}^{(1)} := \{b_1 \in \mathcal{H}(\mathcal{Y}, O) \mid \text{there exists } g \in \mathcal{F} \text{ s.t. } b_1(y) = g(x_0, y) \text{ for all } y \in \mathcal{Y}\};$$
$$B_{y_0}^{(2)} := \{b_2 \in \mathcal{H}(\mathcal{X}, O) \mid \text{there exists } g \in \mathcal{F} \text{ s.t. } b_2(x) = g(x, y_0) \text{ for all } x \in \mathcal{X}\}.$$

Let $(u, v) \in \mathcal{X} \times \mathcal{Y}$ be the first query point that Algorithm 1 makes, we consider $b_1 \in B_u^{(1)}$ and $b_2 \in B_v^{(2)}$.

We first prove the claim holds for deterministic heuristics by induction. This is done by induction on the size of search space $N = |\mathcal{X}|$. Suppose for any two deterministic algorithms $A, B$ on $\mathcal{F}$.

If $N = 1$, then it means that $\mathcal{X} \times \mathcal{Y}$ only consists of one unique NE and $\mathcal{F}$ consists of one payoff function $g$. So for any two deterministic algorithms $A, B$, $r(A, \mathcal{F}) = r(B, \mathcal{F}) = 1$.

Next, assume $N \geq 2$ and $r(A, \mathcal{G}) = r(B, \mathcal{G})$ where $\mathcal{G}$ is any set of payoff functions with unique NE and search space in payoff functions of size $N - 1$ and $\mathcal{G}$ is closed under permutation. Moreover, we assume $\mathcal{F}$ now consists of payoff functions with unique NE and search space in payoff functions of size $N$. We expand out the average of $r(A, \mathcal{F})$.

$$r(A, \mathcal{F}) = \sum_{g \in \mathcal{F}} \Pr(g \text{ is selected from } \mathcal{F}) \cdot T(A, g)$$

If we are lucky, then the first point $(u, v) \in \mathcal{X} \times \mathcal{Y}$ we query is the optimum (opt.) and $(x^*, y^*)$ is the NE of the selected $g$.

$$= 1 \cdot \Pr\left((u, v) \text{ is the opt. s.t. } u = x^* \text{ or } v = y^* \right)$$
$$+ \Pr\left((u, v) \text{ is not the opt. s.t. } u = x^* \text{ or } v = y^* \right)$$
$$\times \left(1 + r\left(A, \mathcal{F}(u, v), (b_1, b_2)\right)\right)$$

Notice that after we query $(u, v)$, we can reduce the whole problem class $\mathcal{F}$ to $\mathcal{F}((u, v), (b_1, b_2))$ with case $N - 1$ where $b_1, b_2$ are defined above.

Lemma 1 shows that if $\mathcal{F}$ is closed under permutation, then $\mathcal{F}((u, v), (b_1, b_2))$ is closed under permutation. Corollary 1 shows that if we restrict $g \in \mathcal{F}((u, v), (b_1, b_2))$, then all the restriction of $g$ on $(\mathcal{X} \setminus \{u\}) \times (\mathcal{Y} \setminus \{v\})$, denoted by $g|_{(\mathcal{X} \setminus \{u\}) \times (\mathcal{Y} \setminus \{v\})}$, exhibits the same unique Nash Equilibrium $(x^*, y^*)$ as $g$. So $\mathcal{F}((u, v), (b_1, b_2))$ is a set of payoff functions with unique NE $(x^*, y^*)$ and the search space of size $N - 1$ and closed under permutation for all $(u, v)$ in which $(u, v)$ is not the opt. s.t. $u = x^*$ or $v = y^*$.

With the help of Corollary 1 and Lemma 1, we apply the inductive hypothesis to this sub-problem class $\mathcal{F}((u, v), (b_1, b_2))$ [7]. Also note that we consider the uniform distribution on problem class $\mathcal{F}$ and thus for any $(u', v') \neq (u, v)$, we have

$$\Pr\left((u, v) \text{ is the opt. s.t. } u = x^* \text{ or } v = y^* \right) = \Pr\left((u', v') \text{ is the opt. s.t. } u = x^* \text{ or } v = y^* \right).$$

Using the inductive hypothesis on the case $N - 1$ with Lemma 2 gives since two sub-problem classes are essentially isomorphic and thus have the same average query complexity from the inductive hypothesis step.

$$r\left(A, \mathcal{F}((u, v), (b_1, b_2))\right) = r\left(B, \mathcal{F}((u', v'), (b_1, b_2))\right).$$

Thus, we can conclude that $r(A, \mathcal{F}) = r(B, \mathcal{F})$ for any deterministic algorithms $A, B$.

---

[7]We defer further explanations of the role of Corollary 1 and Lemma 1 to the supplementary material Section D.

Now, we generalise the result to randomised algorithms. Let $m$ denote the number of different deterministic search heuristics. We consider a randomised search strategy to be a probability distribution $p = (p_1, \cdots, p_m)$ and choose the $i$-th deterministic search strategy with probability $p_i$. It is well-known (see detail in [29]) that the expected cost of a randomised search heuristic is the weighted average of the cost of the deterministic search heuristics. Since all deterministic search heuristics have the same cost, this should also hold for all randomised search strategies. $\square$

**Theorem 4.1.** *Let $\mathcal{X}$ and $\mathcal{Y}$ be any finite sets. Assume that $B \subset \mathbb{R}$ with $k := |B| \geq 2$. Consider any class of two-player zero-sum games $\mathcal{G} \subset \{g : \mathcal{X} \times \mathcal{Y} \to B\}$ such that for all $(x,y) \in \mathcal{X} \times \mathcal{Y}$, there exists a game $g_{x,y} \in \mathcal{G}$ which has $(x,y)$ as unique, pure Nash Equilibrium. Then, the class $\mathcal{G}$ has black box complexity at least $\lceil \log_k |\mathcal{X} \times \mathcal{Y}| \rceil - 1$.*

*Proof of Theorem 4.1.* The proof which uses Yao's minimax principle is analogous to the proof of Theorem 2 in [13]. We need to construct a suitable probability distribution $p$ over the set of games. By assumption, for each $(x,y) \in \mathcal{X} \times \mathcal{Y}$, we can associate one game $g_{x,y}$ which has $(x,y)$ as the unique pure NE. We let $p$ be the uniform distribution over the set of the games $\{g_{x,y} \mid (x,y) \in \mathcal{X} \times \mathcal{Y}\} \subseteq \mathcal{G}$.

We now consider the runtime of any deterministic black-box algorithm $A \in \mathcal{A}_{\text{det}}$ with respect to a random game $g_{x^*,y^*}$ which is sampled according to distribution $p$. The algorithm is a decision tree, where each node is a pair $(x,y) \in \mathcal{X} \times \mathcal{Y}$ corresponding to a query made by algorithm $A$, and each edge corresponds to one of at most $k$ possible outcomes $g(x,y)$ of this query. The runtime of algorithm $A$ on the random game $g_{x^*,y^*}$ corresponds to the depth of $(x^*,y^*)$ in this decision tree. The expected depth is therefore lower bounded by $\lceil \log_k(|\mathcal{X} \times \mathcal{Y}|) \rceil - 1$.

Hence, we have for all deterministic algorithms $A \in \mathcal{A}_{\text{det}}$,

$$\mathrm{E}(T(I_p, A)) \geq \log_k(|\mathcal{X} \times \mathcal{Y}|) - 1.$$

This implies that

$$\min_{A \in \mathcal{A}_{\text{det}}} \mathrm{E}(T(I_p, A)) \geq \log_k(|\mathcal{X} \times \mathcal{Y}|) - 1.$$

By Yao's Principle, for any distribution $q$ over the set of deterministic algorithms $\mathcal{A}_{\text{det}}$,

$$\max_{I \in \mathcal{I}} \mathrm{E}(T(I, A_q)) \geq \min_{A \in \mathcal{A}_{\text{det}}} \mathrm{E}(T(I_p, A))$$
$$\geq \log_k(|\mathcal{X} \times \mathcal{Y}|) - 1.$$

The proof now follows by noting that any randomised algorithm can be described as sampling a deterministic algorithm according some distribution $q$, and applying this algorithm. $\square$

**Theorem 4.2.** *Let $\mathcal{A}$ be the set of all randomised algorithms defined by Algorithm 1 and $T(\mathcal{A}, P)$[8] denote the query complexity of $\mathcal{A}$ with respect to the input payoff matrix $P$ for a two-player zero-sum game. Then, there exists an input matrix $P \in \mathbb{R}^{n \times n}$ with a unique pure Nash equilibrium $(x^*, y^*)$ such that $\mathrm{E}(T(\mathcal{A}, P)) \geq (n+1)/2$. Thus, the black-box complexity with respect to $\mathcal{A}$ of the problem class consisting of all bimatrix games with a unique Nash Equilibrium is at least $(n+1)/2$.*

*Proof of Theorem 4.2.* Given payoff matrix $P$, recall that for any $A \in \mathcal{A}$,

$$T_{\text{LB}}(A, P) := \inf\{C_t > 0 \mid x_t = x^* \text{ or } y_t = y^*\};$$
$$T(A, P) := \inf\{C_t > 0 \mid x_t = x^* \text{ and } y_t = y^*\};$$
$$\mathcal{M} := \{P \in \mathbb{R}^{n \times n} \text{ with a unique PSNE}\}.$$

Clearly, $T(A, P) \geq T_{\text{LB}}(A, P)$. We denote $\mathcal{M}$ as the set of input instances. Now, we estimate the lower bound by using Yao's minimax principle (we denote the query complexity of a specified algorithm namely ALG searching the PSNE of $P$ by $T(\text{ALG}, P)$ and the set of all deterministic black-box adversarial optimisation algorithms by $\mathcal{A}_{\text{det}}$): for any randomised algorithm $A \in \mathcal{A}$,

$$\max_{P \in \mathcal{M}} \mathrm{E}(T(A, P)) \geq \min_{\text{ALG} \in \mathcal{A}_{\text{det}}} \mathop{\mathrm{E}}_{P \sim \text{Unif}(\mathcal{M})} (T(\text{ALG}, P))$$

---

[8]Note that $T(A, P) = T(A, g)$ where $g(x,y) := e_y^T P e_x$ with $e_x, e_y$ denote the elementary probability distribution over probability simplex $\Delta_{\{0,1\}^n}$

Using the definition of $T(\text{ALG}, P)$ and $T_{\text{LB}}(\text{ALG}, P)$ gives

$$\geq \min_{\text{ALG} \in \mathcal{A}_{\text{det}}} \underset{P \sim \text{Unif}(\mathcal{M})}{\text{E}} (T_{\text{LB}}(\text{ALG}, P))$$

Using Theorem 3.1, we know the expected performance of all algorithms on any all problem instance in $\mathcal{M}$ is the same with respect to the unique Nash Equilibrium solution concept. So, we define a new algorithm $\text{ALG}^*$ such that it is a deterministic algorithm starting from $i = j = 1$, and it makes one query in each iteration. It continues to query $P_{1,1}, P_{2,2}, \cdots P_{n,n}$ (i.e. query the entries in the diagonal of the payoff matrix $P$). $\text{ALG}^*$ continues the processes until it reaches either the column or the row of the position of the unique Nash Equilibrium. So, we derive

$$= \underset{P \sim \text{Unif}(\mathcal{M})}{\text{E}} (T_{\text{LB}}(\text{ALG}^*, P))$$

Note that we consider the uniform distribution on $\mathcal{M}$. It means that the probability that $(x^*, y^*)$ lies in the $j$-th column of payoff matrix $P$ is $1/n$ for all $j \in [n]$. Then, the total expected query complexity is $\sum_{j=1}^n j \frac{1}{n} = \frac{n+1}{2}$. So, we derive

$$\geq \frac{n+1}{2}.$$

This completes the proof. $\qquad\square$

**Theorem 4.3.** *The black-box complexity with respect to Algorithm 1 of the binary voting games with problem size $n \in \mathbb{N}$ and a unique Nash Equilibrium is $e^{\Omega(n)}$.*

*Proof of Theorem 4.3.* Given an arbitrary binary voting game with problem size $n$ defined by a payoff function $g : \{0,1\}^n \times \{0,1\}^n \to \mathbb{R}$, we consider the corresponding payoff matrix $P$. For each party, there are $2^n$ possible strategies encoded by a binary bitstring of length $n$. So $P \in \mathbb{R}^{2^n \times 2^n}$. Using Theorem 4.2 with arbitrary algorithms in the class defined by Algorithm 1 and the payoff matrix $P$ gives us the black-box complexity is at least $\frac{2^n+1}{2} = e^{\Omega(n)}$. $\qquad\square$

**Theorem 4.4.** *Given the game class*

$$\text{DIAGONAL}_n := \{\text{DIAGONAL}_{(u,v)} \mid (u,v) \in \{0,1\}^n \times \{0,1\}^n\},$$

*the black-box complexity with respect to Algorithm 1 of $\text{DIAGONAL}_n$ is $\Theta(n)$.*

*Proof of Theorem 4.4.* We consider an algorithm $A \in \mathcal{A}$, which operates in two phases. In each iteration of Phase 1, the algorithm samples $x$ and $y$ uniformly at random, then flips one bit uniformly at random in $x$ to obtain $x'$. It then compares the values of $\text{DIAGONAL}_{u,v}(x, y)$ and $\text{DIAGONAL}_{u,v}(x', y)$. Phase 1 ends when $\text{DIAGONAL}_{u,v}(x, y) \neq \text{DIAGONAL}_{u,v}(x', y)$. Once Phase 1 ends, the algorithm knows that $|u \oplus x|_1 = |v \oplus y|_1$.

In Phase 2, Algorithm $A$ sample a bitstring $y'$ by flipping a single bit in $y$. If $\text{DIAGONAL}_{u,v}(x, y') = -1$, we set $y = y'$, otherwise we repeatedly sampling a new $y'$. Once we get payoff $-1$, we accept the new search point and then sample $x'$ by flipping a single bit in $x$ until repeat the process until we get payoff $1$. This procedure continues until the termination criterion is met.

We now consider how long it takes for Phase 1 to finish (i.e. the search point arrives at the diagonal). Since we are using uniformly at random initialisation, so $Z_1 := |u \oplus x|_1$ and $Z_2 := |v \oplus x|_1$ are subject to two independent binomial random variables $\text{Bin}(n, 1/2)$. We first estimate $\Pr(Z_1 = Z_2)$. Notice that let $Y := n - Z_2$ and $Y$ is also subject to $\text{Bin}(n, 1 - 1/2) = \text{Bin}(n, 1/2)$. So the event $\{Z_1 = Z_2\}$ is equivalent to $\{Z_1 = Y\} = \{Z_1 + Z_2 = n\}$. Thus, we can derive the following estimate by using Stirling's approximation (i.e. $n! \sim \sqrt{2\pi n}(\frac{n}{e})^n$),

$$\Pr(Z_1 = Z_2) = \binom{2n}{n} 2^{-2n} \sim \frac{1}{\sqrt{\pi n}} 2^{2n} \cdot 2^{-2n} = \frac{1}{\sqrt{\pi n}}.$$

So the expected runtime for finishing Phase 1 is $O(\sqrt{n})$.

Now, we consider how long it takes for Phase 2 to reach the optimum. Notice that for each bit, we need 1 query of the payoff function to determine whether it is the correct bit to flip. We need to flip the correct bits for both $x$ and $y$. Since the maximum Hamming distance to the optimum of $\text{DIAGONAL}$ is bounded by $2n$, then the overall runtime is bounded above by $2n = O(n)$. Adding the expected runtime for both Phases gives $O(n) + O(\sqrt{n}) = O(n)$. Together with Theorem 4.1, we can conclude that the black-box complexity of $\text{DIAGONAL}_n$ is $\Theta(n)$. $\qquad\square$

**Theorem 4.5.** *Given the game class,*

$$\text{PLATEAU}_n := \{\text{PLATEAU}_{(u,v)} \mid (u,v) \in \{0,1\}^n \times \{0,1\}^n\},$$

*the black-box complexity with respect to the class of algorithms defined by Algorithm 1 of* $\text{PLATEAU}_n$ *is* $e^{\Omega(n)}$.

*Proof of Theorem 4.5.* Recall the definition of $\text{PLATEAU}_{u,v}$ for arbitrary $u, v$. We call the set $\{(x, y) \mid ||u \oplus x|_1 - \frac{n}{2}| \leq \frac{\varepsilon n}{2}\}$ $x$-independent, and we call any query to this set $x$-independent. Note that for any $x$-independent query has payoff $\text{PLATEAU}_{u,v}(x, y) = f(v \oplus y)$, i.e., it is independent of $x$. Furthermore, note that the Nash Equilibrium is not $x$-independent.

We will apply Yao's Principle with respect to the distribution $p$ over instances where $u$ is sampled uniformly at random among all bitstrings of length $n$, and $v = 0^n$. We consider the average case runtime of deterministic algorithms with respect to distribution $p$. Such algorithms can be modelled as binary decision trees: in each node, the algorithm makes a query $(x, y)$, and for each outgoing edge, the algorithm obtains one of two possible payoff values $\text{PLATEAU}_{u,v}(x, y) \in \{-1, 1\}$. We call the $x$-independent path in the decision tree the longest path $(x_1, y_1), \ldots, (x_t, y_t)$ such that $(x_1, y_1)$ is the root node and all nodes $(x_i, y_i)$ for $i \in [t]$ are $x$-independent. We let $t$ denote the length of the $x$-independent path. Since the outcome of an $x$-independent query only depends on $v \oplus y$ which is deterministic, the $x$-independent path of a decision tree is unique. Furthermore, the length of the $x$-independent path is a lower bound on the runtime of the algorithm since the Nash Equilibrium is not $x$-independent.

We now lower bound the length of the $x$-independent path. Let $(x_i, y_i)$ be any fixed query in the decision tree, and $F_i$ the event that $(x_i, y_i)$ is $x$-independent. We sample $u$ by picking each bit independently and uniformly at random in $\{0, 1\}$. This implies that the number of 1 bits for each bit is subject to a Bernoulli random variable $\text{Bin}(1/2)$. We therefore have $|u \oplus x_i|_1$ is binomially distributed $\text{Bin}(n, 1/2)$. Applying Chernoff's bound, we get

$$\Pr(F_i) = \Pr\left(||u \oplus x_i|_1 - \frac{n}{2}| > \varepsilon \frac{n}{2}\right) \leq 2e^{-\varepsilon^2 n/6}. \tag{1}$$

Let $E_1$ be the complement of all the failure events, i.e., $E_1 = \overline{\cup_{i=1}^t F_i}$. Therefore, by a union bound, the $x$-independent path has length $t = 2e^{\varepsilon^2 n/12}$ with probability

$$\Pr(E_1) = 1 - \Pr\left(\cup_{i=1}^t F_i\right) \geq 1 - t2e^{-\varepsilon^2 n/6} = 1 - e^{-\Omega(n)}.$$

We then have by the law of total probability

$$\begin{aligned}
\min_{A \in \mathcal{A}} \mathrm{E}(T(I_p, A)) &\geq \min_{A \in \mathcal{A}} \Pr(E_1)\, \mathrm{E}(T(I_p, A) \mid E_1) \\
&\geq (1 - e^{-\Omega(n)})2e^{\varepsilon^2 n/12} \\
&= e^{\Omega(n)}.
\end{aligned}$$

The proof now follows by Yao's minimax Principle. The expected worst-case runtime of any randomised algorithm is lower bounded by the average case runtime of deterministic algorithms.

$$\max_{I \in \mathcal{I}} \mathrm{E}(T(I, A_q)) \geq \min_{A \in \mathcal{A}} \mathrm{E}(T(I_p, A)) = e^{\Omega(n)}.$$

$\square$

# E   Further Explanations for Sub-Problem Class

We provide an example to illustrate the concept of the sub-problem class (see Definition 3).

Figure 2 is an example of a sub-problem class. When querying $(x_1, y_1)$, we also check the corresponding row and column to verify whether the unique Nash Equilibrium lies on this blue cross or not. If the answer is negative, then we restrict to the sub-matrix by removing the blue entries in $P$.

$$P = \begin{pmatrix} & & \\ & ? & \end{pmatrix} \begin{matrix} y_1 \\ \\ y^* \end{matrix}$$
$$x_1 \quad x^*$$

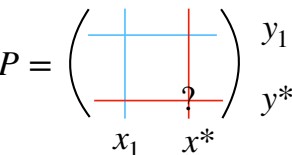

Figure 2: Example of a sub-problem class. Given a payoff matrix $P$, the blue entries mean that the actual values are already assigned to $g(x_1, \cdot)$ and $g(\cdot, y_1)$. $(x^*, y^*)$ is the Nash Equilibrium we search for. The sub-problem class here means that the problem consists of different payoff matrices with given blue entries.

## F  Further Explanations for the role of Corollary 1 and Lemma 1 in Theorem 3.1

In the main proof of Theorem 3.1, we rely on proof by induction: assume the theorem holds for a search space of size $N$, we want to show it also holds for $N + 1$. Corollary 1 and Lemma 1 are crucial since they ensure that for every payoff function $g \in \mathcal{F}$, there is a corresponding function $g'$ where the NE $(x^*, y^*)$ is switched to $(\sigma(x^*), \tau(y^*))$ (i.e. $\sigma, \tau$ are two permutations on $\mathcal{X}, \mathcal{Y}$ respectively) and the NE is still unique in the game defined by $g'$. This means that the inductive hypothesis can be applied to $g'$ due to the closure under permutation. In short, Corollary 1 and Lemma 1 establish a property of the problems or payoff functions under consideration that is preserved under permutations. This property helps to ensure that the induction step can be applied successfully.

## G  Further Explanations for bitwise exclusive or in Black-Box Complexity Analysis

We define $\oplus$ as bitwise exclusive or. $\oplus$ means that for any two binary bitstrings $x = x_1 \cdots x_n$ and $u = u_1 \cdots u_n$, $x \oplus u := (x_1 \oplus u_1, \cdots, x_n \oplus u_n)$ where $1 \oplus 1 = 0, 1 \oplus 0 = 1, 0 \oplus 1 = 1, 1 \oplus 1 = 0$. We introduce bitwise exclusive or here to generate problem instances with the same structure. For example, $\mathrm{DIAGONAL}(x, y)$ is only one possible problem instance in binary voting games and to find its optimum, we only need to design the algorithm querying $(1^n, 1^n)$ to reach its optimum. And its black-box complexity is trivially $\Theta(1)$. Using bitwise exclusive or $\oplus$ (i.e. $\mathrm{DIAGONAL}(u \oplus x, v \oplus y)$ where $(u, v) \in \mathcal{U} \times \mathcal{V}$ is sampled uniformly at random), we can generate a set of problem instances which have the same structure as $\mathrm{DIAGONAL}(x, y)$ and the goal of black-box algorithms is to find $(x, y)$ such that $(u \oplus x, v \oplus y) = (1^n, 1^n)$. (Note that if $(u, v) = (0^n, 0^n)$, then $\mathrm{DIAGONAL}(0^n \oplus x, 0^n \oplus y)$ is reduced to the vanilla $\mathrm{DIAGONAL}(x, y)$.) This bitwise exclusive generator can avoid the trivial black-box complexity mentioned above and reflect the true difficulty of the class $\mathrm{DIAGONAL}$. We refer to [13, 9] for more detail about black-box complexity theory.

