# OpenReview forum: "No Free Lunch Theorem and Black-Box Complexity Analysis for Adversarial Optimisation"
_NeurIPS.cc/2024/Conference — NeurIPS 2024 poster_

### Official Review · Reviewer_RwGp · 2024-06-17

**Soundness:** 3
**Presentation:** 3
**Contribution:** 3
**Rating:** 6
**Confidence:** 3

**Summary:**

This paper considers the complexity of finding (Pure Strategy) Nash Equilibrium in black-box optimization for adversarial optimization.

By denoting the loss of a learned solution $x\in X$ on a possible test case $y\in Y$ as $g(x,y)$, the authors show that no algorithm is on average better than any other in finding a PSNE of $g\in \mathcal F$, given that $\mathcal F$ is closed under permutation, via constructing an isomorphism result which roughly says that if the algorithm doesn't at the first time query the unique PSNE $(x^\ast,y^\ast)$, then every possible sub-problem is isomorphic to another. This NFL result distinguishes finding a NE from finding an "always optimal" or "worst-case optimal" solution.

The authors then derive the query complexity lower bounds for general black-box adversarial optimization and two-player zero-sum bimatrix games, and also their implications to some two-player zero-sum games.

**Strengths:**

1. This NFL-style result distinguishes finding NE from finding an "always optimal" or "worst-case optimal" solution, which is important towards understanding the various solution concepts in adversarial optimization.
2. Via the NFL-style result, various query complexity lower bounds are also derived for adversarial optimization and two-player games.

**Weaknesses:**

1. This paper assumes a unique PSNE in $g$, but it may not be this case in ML (e.g., $X$ corresponds to a set of over-parameterized NN and $y$ is some train/test loss).
1. While the main text looks pretty technically rigorous, it is a little hard to read. For example, in a sub-problem definition, what does the $b_1,b_2$ usually mean and how are they constructed? The answer to this question doesn't seem to appear anywhere in the main text as well.

**Questions:**

1. In the proof of Theorem 3.1, the authors mentioned "$b_1$ and $b_2$ are defined in Definition 6". Is this really correct? I don't feel Definition 6 infers the construction of $b_1$ and $b_2$.
2. In Theorem 4.1, the condition that "each query has at most k ≥ 2 possible answers" seems pretty strong -- what if the noise on $g(x,y)$ is a continuous r.v.?
3. Can you discuss more regarding the "always optimal" solution concept in Definition 2? Is it likely to exist for a general adversarial optimization problem? If not, why did the original authors consider it?

**Limitations:**

Included in sec. 5

---

> ### Author Rebuttal · Authors · 2024-08-06
>
> **RwGpW2. While the main text looks pretty technically rigorous, it is a little hard to read ...**
>
> A: We agree with the reviewer that the role of $b_1$ and $b_2$ was not fully explained.
>  We will improve this in the final version. In particular, Definition 5 can be improved as follows:
>
> Let $O\subseteq \mathbb{R}$.
> Let $\mathcal{F}$ be any subset of the set of payoff functions $g:\mathcal{X} \times \mathcal{Y} \rightarrow O$ with a unique Nash Equilibrium and such that $\mathcal{F}$ is closed under permutation.    For any given $(x_1,y_1) \in \mathcal{X} \times \mathcal{Y} $, any function $b_1:\mathcal{Y} \rightarrow  O$, and any function $b_2:\mathcal{X} \rightarrow O$,
>     we define a sub-problem class $\mathcal{F} \left((x_1,y_1),(b_1,b_2) \right)$ with respect to $\mathcal{F}$ as follows:
>     $f \in \mathcal{F} \left((x_1,y_1),(b_1,b_2) \right)$ iff there exists a $g \in \mathcal{F}$ such that
>
> (1)  $g(x_1,y)=b_1(y)$ for all $y \in \mathcal{Y}$;
>
> (2) $g(x,y_1)=b_2(x)$ for all $x \in \mathcal{X}$;
>
> (3) $f$ is a restrction of $g$ on $\left(\mathcal{X} \setminus \\{x_1\\} \right) \times \left( \mathcal{Y} \setminus \\{y_1\\} \right)$.
>
> The formal definition of restriction/extension can be found in the response of PuqfQ1.
>
>  In terms of the construction of $b_1, b_2$, we explain in detail as follows.
>   To get the intuition of what $b_1,b_2$ means in our analysis, we extend the discussion of **Section E** here.
>   As shown in **Figure 2, Section E, Appendix**, after querying $(x_1,y_1)$, both strategies of $x_1,y_1$ are known.
>   We define $b_1(y):=g(x_1,y)$ for all $y \in \mathcal{Y}$ and $b_2(x):=g(x,y_1)$ for all $x \in \mathcal{X}$.
>   Note that the definition of $b_1, b_2$ depends on $g \in \mathcal{F}$.
>   We consider a black-box model where the algorithm learns all the blue entries in payoff matrix $P$ (Figure 2) when querying the entry ($x_1,y_1)$ (i.e., row $x_1$ and column $y_1$).
>   If there is no Nash equilibrium among the blue entries, we exclude this row and column in the payoff matrix and restrict the problem to a smaller sub-problem.
>   Now, we define $b_1,b_2$ formally:
>
>
>   We denote the set of all functions from a set $\mathcal{X}$ to a set $O$ by $\mathcal{H}(\mathcal{X},O)$ and the set of all well-defined functions from a set $\mathcal{Y}$ to a set $O$ by $\mathcal{H}(\mathcal{Y},O)$.
>
>  Given a subset $\mathcal{F}$ of all the payoff functions $g: \mathcal{X} \times \mathcal{Y} \rightarrow O$ with a unique NE where $\mathcal{F}$ is closed under permutation. For all $(x_0,y_0) \in \mathcal{X} \times \mathcal{Y}$,
>   define
>
>   \begin{align*}
>         B_{x_0}^{(1)}:&=
>  \\{b_1 \in \mathcal{H}(\mathcal{Y},O) \mid \text{there exists $g \in \mathcal{F}$ s.t. } b_1(y)=g(x_0,y) \text{ for all $y \in \mathcal{Y}$} \\}; \\\\
>          B_{y_0}^{(2)}:&=
>   \\{b_2 \in \mathcal{H}(\mathcal{X},O) \mid \text{there exists $g \in \mathcal{F}$ s.t. } b_2(x)=g(x,y_0) \text{ for all $x \in \mathcal{X}$}\\}.
>   \end{align*}
>
>   Let $(x_1,y_1) \in \mathcal{X} \times \mathcal{Y}$ be the first query point that Algorithm 1 makes,
>   we consider $b_1 \in B_{x_1}^{(1)}$ and $b_2 \in B_{y_1}^{(2)}$.
>   Now, they should be well-defined in the proof of Theorem 3.1.
>
>  We will include this explanation in the proof to improve the accessibility of our paper.
>
> **RwGpQ1. In the proof of Theorem 3.1, the authors mentioned, "b1, b2 are defined in Definition
> 6". Is this really correct? I don’t feel Definition 6 infers the construction of b1 and b2.**
>
> A: Yes, you are right. There is a typo; it should be Definition 5 rather than 6. We will correct
> this in the updated version. The construction part can be checked in the response to RwGpW2.
>
> **RwGpQ3. Can you discuss more regarding the "always optimal" solution concept in Definition
> 2? Is it likely to exist for a general adversarial optimization problem? If not, why did the
> original authors consider it?**
>
> A: Yes, we agree with the reviewer that an "always optimal" solution can hardly exist for a
> general adversarial optimisation problem. Even in their previous paper, Service and Tauritz [12]
> acknowledge this by stating:
>
> “in many real-world problems, there are no candidate solutions which perform best over all possible
> test cases. That is, there is often a trade-off between performance on test cases.”
>
> We believe they consider this “ideal solution concept" to theoretically demonstrate the existence of a
> "free lunch" with respect to it. While their result and the result of Wolpert and Macready [14] can be
> considered worthwhile contributions to No Free Lunch (NFL) in adversarial optimisation, we argue
> that it is necessary to consider other solution concepts.
>
> In fact, we are not satisfied with these "free lunch" results since an "always optimal" solution rarely
> exists in real-world applications, and such a "free lunch" result might sometimes be misleading. It is
> important to recognise that every approach, including coevolutionary approaches or other black-box
> adversarial optimisation algorithms have limitations. This is one of the key messages of our paper.
>
> We will extend our discussion in the updated version by including the response above.

---

> > ### Comment · Reviewer_RwGp · 2024-08-12
> >
> > Thanks for your detailed responses. I appreciate that.

---

### Official Review · Reviewer_7LPo · 2024-07-12

**Soundness:** 3
**Presentation:** 3
**Contribution:** 3
**Rating:** 6
**Confidence:** 3

**Summary:**

In this paper, the authors theoretically analyze the query complexity for general black-box Adversarial Optimisation under the "closed under permutation" assumption.  A no-free-lunch theorem is proved for all algorithms in achieving the same average performance of all possible problem instances (or problem instances c.u.p.) with a unique Nash Equilibrium in a two-player zero-sum game setting.  Moreover, the authors provide Lower Bounds for General Black-Box Adversarial Optimisation (with possible answers assumption ) and Two-player Zero-Sum Bimatrix Games, respectively.

**Strengths:**

1.  The paper is well-written and well-organized.


2.  The theoretical results are general to cover many interesting adversarial optimization cases.  The no-free-lunch theorem shows the difficulty of optimizing the whole family of problem instances in two-player zero-sum games.   The lower bound of query complexity for  General Black-Box Adversarial Optimisation (with possible answers assumption for each query) and Two-player Zero-Sum Bimatrix Games are also interesting.

**Weaknesses:**

1. The technical challenge of the analysis for black-box adversarial optimization over the standard adversarial optimization analysis is not clearly discussed.   What the key technique employed for the black-box cases compared with the standard  Nash Equilibrium analysis for two-player zero-sum games is not clear.


2. The theoretical analysis is restricted to finite (discrete) problems instead of continuous problems.  This restriction may limit the possible application of the theoretical results.

**Questions:**

1.  What is the key technical challenge for the theoretical analysis of black-box adversarial optimization compared with white-box adversarial optimization?


2. What is the key technique employed in this paper to solve the challenge compared with the standard  Nash Equilibrium analysis for two-player zero-sum games or adversarial optimization analysis?

**Limitations:**

N.A.

---

> ### Author Rebuttal · Authors · 2024-08-06
>
> **7LPoW1+Q1.**
> - **The technical challenge of the analysis for black-box adversarial optimization
> vs the standard adversarial optimization analysis**
> - **The key technique employed for the black-box cases for two-player zero-sum games**
> - **Black-box adversarial optimisation vs white-box adversarial optimisation**
>
>
> A: The detailed discussion of key technical contributions,  challenges and technique can be found in
> **Sections 1.2 and 1.3**.
>
> There are different definitions of white-box in the literature. To answer
> the question, it is necessary to define the meaning of white-box adversarial optimisation. For
> comparison, we need to define white-box adversarial optimisation. We have defined black-box
> adversarial optimisation in **lines 23-30** and **Algorithm 1**. Since the reviewer did not specify which
> interpretation they had in mind precisely, we consider two possible interpretations:
>
> (1) **White-box adversarial optimisation with full access to the payoff function:** This means the
> payoff matrix in the two-player zero-sum game is given and known. In this case, finding the Nash
> Equilibrium (NE) can be formulated as a linear programming problem, which can be solved in
> polynomial time using algorithms such as the ellipsoid method or interior-point method [2, 9].
>
> (2)**White-box adversarial optimisation with gradient access:** The gradient is accessible if the
> payoff function is differentiable. A well-known method for solving the saddle-point problem is
> gradient descent ascent (GDA) and its variants, including stochastic/optimistic GDA, which have
> been extensively studied [8, 4].
>
> Compared with these two types of white-box adversarial optimisation, black-box adversarial optimi-
> sation considered in this paper only allows querying the payoff function in each iteration, and the
> payoff function is usually not explicitly known. The tools used in these two white-box cases are not
> applicable in our scenario. This is why we employ tools from game theory **(see lines 66-69)** and
> introduce black-box complexity to analyse the black-box adversarial model **(see lines 70-74)**.
>
>
> **7LPoQ2. The key technique employed in this paper to solve the challenge compared
> with the standard Nash Equilibrium analysis for two-player zero-sum games or adversarial
> optimization analysis**
>
> A: For the standard Nash Equilibrium analysis for two-player zero-sum games, the payoff
> function is explicitly known. As mentioned in our response, the tools used in such cases are not
> applicable here. We have also discussed other works, such as the computational complexity of
> computing NE in general-sum games being PPAD-complete and the complexity of fictitious play in
> two-player potential games with a unique NE, in **lines 568-583**.
>
> The key technique employed in our paper involves using game-theoretic tools and Yao’s principle to
> analyse black-box adversarial models. This approach is necessary because, unlike in standard NE
> analysis, we only have access to the payoff function through queries, and the function is not explicitly
> known and discrete (hence, no gradient is accessible).

---

> > ### Comment · Reviewer_7LPo · 2024-08-13
> >
> > Thanks for the authors' detailed response.  My concern has been well addressed.  I have no further questions.

---

### Official Review · Reviewer_Puqf · 2024-07-13

**Soundness:** 3
**Presentation:** 3
**Contribution:** 3
**Rating:** 5
**Confidence:** 4

**Summary:**

Black-box optimization is a critical area in the field of optimization. The original No Free Lunch (NFL) theorem highlights the inherent limitations of traditional black-box optimization and learning algorithms, establishing a theoretical basis for these methods. The paper addresses a long-standing problem in NFL analysis for adversarial (maximin) optimization. The authors try to prove a new NFL theorem for general black-box adversarial optimization, specifically when Nash Equilibrium (NE) is used as the solution concept. This implies that when NE is the goal, the average performance of all black-box adversarial optimization algorithms is equivalent. Using Yao’s maximin principle and the new NFL theorem, the paper provides general lower bounds for the query complexity required to find Nash Equilibrium in adversarial optimization.

The authors prove the theoretical impossibility of a universally effective adversarial optimization algorithm and highlight the impact of solution concept selection. It introduces black-box complexity to assess the difficulty of learning the unique optimum and solving two-player zero-sum games, laying the foundation for future research on adversarial optimization.

**Strengths:**

S1. The paper demonstrates a high level of theoretical rigor in proving an NFL Theorem for general black-box adversarial optimization, specifically focusing on Nash Equilibrium as the solution concept. The proofs and technical details are sufficiently detailed.

S2. The paper effectively formulates the problem statement by emphasizing the importance of solution concepts in adversarial optimization. The paper contributes by addressing adversarial optimization and providing a new perspective on the limitations of black-box algorithms in this context.

**Weaknesses:**

W1. The notations used in the paper are not clear in some places. For example, in line 92, v is introduced as a real number. Then for i in [1,n], what is v_i in line 93? In the double-line equation between line 149 and line 150, the notation is not clear. Terms like 'extension' and 'restriction' of a function should ideally be defined earlier in the paper. Here, it is only somewhat discussed in the appendix.

W2. The paper seems to focus more on theoretical analysis and proofs, lacking empirical validation or practical demonstrations of the proposed concepts and findings. Incorporating experimental results could enhance the credibility and applicability of the research.

W3. For the theoretical analysis discussed in the paper, there is only one practical application provided viz., two-player zero-sum games with a unique Nash Equilibrium (NE). The authors introduce certain complexity in this game by considering plateaus. However, the application is still a trivial one and a study on a broader range of adversarial optimization problems with different solution concepts would have been insightful.

W4. The paper cannot rely on the appendix. The text should be self-contained. The appendix contains many crucial information needed for understanding and judging the paper. I didn't particularly like the current structure of the paper, where related work has ended up in the appendix.

W5. In the checklist, the authors mention that from a practical viewpoint, more careful benchmark selections are suggested for use in many black-box optimization applications that solve maximin problems with complicated constraints. However, an elaborate discussion on the practical applicability is not discussed in the paper. Such a discussion would have been insightful.

**Questions:**

Please clarify the points in the Weaknesses section.

**Limitations:**

Yes, the limitations of the proposed method are discussed in the paper.

---

> ### Author Rebuttal · Authors · 2024-08-06
>
> **PuqfW1. The notations used in the paper are not clear in some places.**
>
> A: We will improve the accessibility of this paper in the updated version.
>     There is a typo here, $v \in \mathbb{R}^n$ rather than  $v \in \mathbb{R}$ and $v_i$ is the $i$-th opponent of $v$.
>     We refer to $g$ as the extension of $g'$ in line 149 and 150 and it is defined in (1).
>     Note that restrictions and extension are two well-established mathematical terms.
>     We define them separately here and will include these formal definitions in the preliminaries later.
>
> Def (restriction):
>  Let $f : X \to Y$  be a function from a set $X$ to a set $Y$.
> If $A$ is a subset of  $X$, then the **restriction** of $f$ to $A$ is the function:
> \begin{align*}
> f|_A : A &\to Y \\\\
> x &\mapsto f(x).
> \end{align*}
>
> Def (extension):
> Let $f : X \to Y$ be a function and $A$ and $B$ be sets such that $X \subseteq A$ and $Y \subseteq B$.
> An **extension** of $f$ to $A$ is a function $g : A \to B$ such that $f(x) = g(x)$ for all  $x \in X$.
>  Alternatively, $g$ is an **extension** of $f$ to $A$ if $f$ is the restriction of $g$ to $X$.
>
> **PuqfW4. The paper cannot rely on the appendix. The text should be self-contained.**
>
> A:  We agree that the appendix is not the appropriate place for related work. In the revised
> version of the paper, we will move the extra related work from the appendix to the main text where
> necessary and compress the main text. However, it is not feasible to include all proof details in the
> main body of any comprehensive theory paper. As is traditional for theoretical papers in AI/ML
> conferences, detailed proofs are deferred to the appendix (e.g., see [11, 5, 3, 4]). We will include
> proof sketches or explanations in the main text, as seen in line 183, to ensure the paper remains
> self-contained.
>
> **PuqfW5. In the checklist, the authors mention that from a practical viewpoint, ...
> However, an elaborate discussion on the practical applicability is not discussed in the paper.**
>
> A: As a theory paper, we focus on rigorous proof and analysis of general adversarial
> optimisation problems. For the discussion of careful benchmark selections in black-box adversarial
> optimisation, we have used binary voting games (two-player zero-sum games) as a practical
> illustration of BBC analysis. The detailed discussion of the practical applicability of black-box
> complexity analysis can also be checked in **Section 4.4.3**.

---

> > ### Comment · Reviewer_Puqf · 2024-08-14
> >
> > I thank the authors for their detailed response. I understand that this work is mostly theoretical and this paper has the potential to set the foundation in this domain. However, I am still concerned that: (a) the main text is hard to read (which is agreed upon by Reviewer RwGp) + there are some discrepancies in the notations and (b) practical applicability is not sufficiently discussed (even in Section 4.4.3, as suggested by the authors). So, I have decided to keep my score.

---

> > > ### Author Response · Authors · 2024-08-14
> > >
> > > Thank you for your response and for acknowledging the contribution of our work in laying the foundation for black-box adversarial optimisation. We provide a brief explanation here:
> > >
> > > **(1) Readability and Notation Discrepancies:**
> > >
> > > Precise mathematical definitions and notations are essential for rigorous theoretical research. However, we also understand the importance of clarity and accessibility. To address this, we will put extra emphasis on accessibility in the revised version, including more explanations in plain English and ensuring consistent use of notations throughout the text.
> > >
> > > We have already **clarified** definitions as requested by Reviewer **RwGp** and included explanations of well-established terms like "restriction" and "extension," as suggested by Reviewer Puqf.
> > >
> > > These efforts will be extended in the revised version to make the paper more accessible without compromising its rigour.
> > >
> > > **(2) Practical Applicability:**
> > >
> > > As noted in our global response to Q3, our results focus on theoretical impossibility results, which in nature usually do not have immediate practical applications.
> > >
> > > However, this **does not lower** their value  (see NFL for traditional black-box optimisation [13] or Arrow's impossibility theorem [15]). As a key message of our paper, theoretical impossibility results are crucial for understanding the limitations of black-box adversarial optimisation.
> > >
> > > We have discussed the usage of black-box complexity analysis in Section 4.4.3 and will include the discussion of theoretical impossibility results in our revised version as well. Future work could explore the potential practical implications of these theoretical findings, which might eventually inform the design and evaluation of black-box optimisation algorithms in applied settings like binary voting games discussed in this paper.
> > >
> > >
> > >
> > > Above all, we will include all the discussion in our revised version.  Moreover, we are pleased to note that the other reviewers are **satisfied** with our responses and responded **positively** with no further questions.  We appreciate your detailed feedback and hope our explanations help resolve your concerns.
> > >
> > >
> > > **Reference**
> > >
> > > [13] D.H. Wolpert, W.G. Macready. No Free Lunch Theorems for Optimization. TEVC, 1997.
> > >
> > > [15] Kenneth J Arrow. A difficulty in the concept of social welfare. Journal of political economy, 1950.

---

> ### Comment · Area_Chair_SHHD · 2024-08-12
>
> Reviewer Puqf,
>
> Since you gave a borderline score for this paper, please engage in discussions with the authors and see if the rebuttal addressed your concerns.
>
> Thanks,
>
> AC

---

### Official Review · Reviewer_Xe7y · 2024-07-15

**Soundness:** 2
**Presentation:** 3
**Contribution:** 2
**Rating:** 5
**Confidence:** 3

**Summary:**

The paper mainly focuses on the analysis of black-box adversarial optimization algorithms with an emphasis on Nash Equilibrium (NE) as the solution concept. It introduces the concept of No Free Lunch (NFL) Theorem for general black-box adversarial optimization, showcasing the equal average performance of all algorithms when NE is the solution concept. The paper also delves into the black-box complexity analysis and provides lower bounds for query complexity in finding NE. Overall, it contributes to the understanding of the limitations and performance evaluation of black-box adversarial optimization algorithms under the Nash Equilibrium solution concept.

**Strengths:**

This paper provides rigorous analysis and theoretical foundation concerning black-box adversarial optimization algorithms, particularly focusing on the Nash Equilibrium as the solution concept. The paper provides a novel proof of the No Free Lunch Theorem in the context of adversarial optimization, shedding light on the equal performance of algorithms when NE is considered. Additionally, the introduction of black-box complexity analysis adds depth to understanding the query complexity in finding NE. Overall, the paper's strengths include its theoretical contributions, clarity in presentation, and insightful analysis of black-box adversarial optimization in the context of NE.

**Weaknesses:**

See questions below.

**Questions:**

1. This paper is theoretically comprehensive. However, providing simple empirical verification on binary voting might increase the credibility of the theoretical founding. I am wondering if it is possible to conduct experiments to verify the proposed lower bound.
2. I am curious about more difficult games than DIAGONAL and PLATEAU that can be solved by Algorithm 1. Can you provide other examples?
3. Algorithm 1 is still a random search algorithm. Can it be generalized to some real-world games? Please discuss more on different classes of search heuristics rather than random search. How will the results of black-box complexity change?

**Limitations:**

No potential negative societal impact.

---

> ### Author Rebuttal · Authors · 2024-08-06
>
> **RXe7yQ2 I am curious about more difficult games than DIAGONAL and PLATEAU that can
> be solved by Algorithm 1. Can you provide other examples?**
>
> A: Before answering the question, we would like to clarify two confusions in the question:
>
> (1) As mentioned in the paper **(lines 223-230)**, Algorithm 1 represents a broad class of algorithms.
> Therefore, we cannot claim that all instances of Algorithm 1 solve Diagonal and Plateau.
>
> (2) While all instances of the Diagonal problem class can be solved by some instance of Algorithm 1
> in polynomial runtime, Plateau is already a challenging problem in terms of polynomial solvability. In
> particular, we prove that all instances of Algorithm 1 need exponential time to solve Plateau, making
> it inefficient. Hence, we do not consider it significant to find more difficult games than Plateau.
>
> (3) The reason we chose these examples is to illustrate two kinds of problems within the general class of binary voting games with unique NE: the **polynomial-solvable** class (i.e., there exists an algorithm $A \in \mathcal{A}$ that can solve all problem instances of this class in polynomial runtime) and the **non-polynomial-solvable** class (i.e., there exists **no** algorithm $A \in \mathcal{A}$ that can solve all problem instances of this class in polynomial runtime).
>   As mentioned in **Section 4.4.3**, these examples and the general class of binary voting games with unique NE are sufficient to illustrate the use of black-box complexity in black-box adversarial optimisation.
>
> **RXe7yQ3 Algorithm 1 is still a random search algorithm. Can it be generalised to some real-
> world games? Please discuss more on different classes of search heuristics rather than random
> search. How will the results of black-box complexity change?**
>
> A: Before answering the question, we would like to point out another confusion in the
> question:
>
> Although the algorithm(s) make random decisions, it does not make it a pure random
> search (sampling solutions uniformly at random).
> We would like to clarify that Algorithm 1 is **not** a simple class of random search algorithms but defines a general class of algorithms.  As is well-known in the literature, clever use of
> randomness can lead to simpler and more robust algorithms.
>
> We extend our discussion from the original paper **(lines 223-233)** here. Algorithm 1 defines various
> black-box adversarial algorithms, including coevolutionary heuristics [7 , 6, 1], bandit learning
> algorithms [10 , 3], and other black-box query-only randomised algorithms like FINDPSNE (designed
> to learn the NE in bimatrix games by querying the payoff matrix) [9], by specifying different
> probability distributions PI(t) in Line 3 of Algorithm 1.
>
> The main aim of Section 4 is to provide a general performance measure for such a broad class of black-box models.
> For other algorithms outside the class defined by Algorithm 1, if in the binary voting games they act like a decision tree
> algorithm, we conjecture that the black-box complexity (BBC) will remain the same. If not, we may
> need to examine different classes on a case-by-case basis.

---

> > ### Comment · Reviewer_Xe7y · 2024-08-13
> > **Acknowledgement of reading the authors' responses**
> >
> > I would like to thank the authors for clarifying my doubts. The responses provided by the authors adequately addressed my concerns.

---

> ### Comment · Area_Chair_SHHD · 2024-08-12
>
> Reviewer Xe7y,
>
> Since you gave a borderline score for this paper, please engage in discussions with the authors and see if the rebuttal addressed your concerns.
>
> Thanks,
>
> AC

---

### Author Rebuttal · Authors · 2024-08-06

We thank all the reviewers for their useful and detailed comments.

Due to the space limit, we only keep the reference list in the global response. All the responses will use the **same** reference list.

**Q1 Two-player zero-sum game with a unique NE or PSNE. PuqfW3; RwGpW1**

A: Our current theoretical analysis focuses on two-player zero-sum games with a unique Nash
Equilibrium (NE). We agree that exploring other solution concepts and games would be insightful.
However, we argue that our analysis solves a non-trivial open problem.
Choosing this class of games is reasonable for the following reasons:

(1) The No Free Lunch (NFL) theorem for any solution concept is still a long-standing open problem.
One of the main messages of this paper is that, despite prior work showing the existence of a “free
lunch” in adversarial optimisation (e.g., [13]; [12]), we prove a NFL theorem for the unique Nash
Equilibrium. This uncovers the fact that there is no “silver bullet” in adversarial optimisation for
all solution concepts. To our knowledge, this is the first work to answer this question negatively,
and the unique NE solution concept is sufficient to serve this aim.

(2) The technical level is state-of-the-art within the area. As discussed in Section 1.2, we make only
a very weak assumption: limited query access to the payoff function, with no assumptions about
properties such as convexity, continuity, or differentiability. Therefore, we need to introduce tools
from game theory and Yao’s principle.

(3) Two-player zero-sum games with a unique NE are commonly studied in AI/ML literature. For
instance, (detailed discussion can be found in **Section A** in Appendix),

(a) Learning in Games:  for example, see [11], [10], [9].

(b) Co-evolutionary Heuristics: for example, see [7],[6],[1].

In summary, two-player zero-sum games with a unique NE represent a broad and meaningful class of
games, providing a reasonable starting point. While it is indeed interesting to generalise the analysis
to more complex solution concepts such as mixed NEs or broader classes of adversarial optimisation
problems, it is a necessary first step in order to be able to analyse more general classes of games.

**Q2 The theoretical analysis is restricted to finite (discrete) problems. 7LPoW2; RwGpQ2.**

A: We agree that the current result is restricted to finite (discrete) problems and only
considers the deterministic case where "each query has at most k ≥ 2 possible answers".
There are two reasons for using this condition:

(1) For consistency with previous literature and a fair comparison, we use the same setting as the previous
NFL/FL results [13, 14, 12] and emphasise that under the same condition, the solution concept
plays a role in the construction of NFL/FL.

(2) The generalisation of such a result is interesting, however, as shown in the paper, under this
condition, the proof/analysis is still challenging and non-trivial. It is necessary to understand the
basic setting before moving to a more general setting.

In conclusion, despite the limitations, this setting provides a clear and rigorous starting point. Future
work can build on these results to explore continuous and more complex scenarios. As mentioned
in **line 384**, we may need further assumptions or use other tools to derive the NFL analysis for
other scenarios. For a broad class of possible applications on two-player zero-sum games with finite
strategies and unique NE, we refer to our response to Q1.

**Q3 Empirical verification of our theoretical result. RXe7yQ1; PuqfW2**

A: It is impossible to verify impossibility results, such as our black-box results, empirically.
In particular, we are proving an impossibility result for an infinitely large class of algorithms and for an infinitely large class of problems (c.f. Definition 7).
To verify the result empirically, one would
have to try an infinite number of algorithms on an infinite number of problem instances. Clearly, this
is not possible. This paper is theoretical in nature and aims to provide a general understanding of
black-box adversarial optimisation.

**References**

[1] A. Benford, P.K. Lehre. Runtime Analysis of Coevolutionary Algorithms on a Class of Symmetric Zero-Sum Games. In GECCO, 2024.

[2] S. Bubeck et al. Convex Optimization: Algorithms and Complexity. Found. Trends Mach. Learn., 2015.

[3] Y. Cai, H. Luo, C.-Y. Wei, W. Zheng. Uncoupled and Convergent Learning in Two-Player Zero-Sum Markov Games with Bandit Feedback. In NeurIPS, 2023.

[4] C. Daskalakis, I. Panageas. The Limit Points of (Optimistic) Gradient Descent in Min-Max Optimization. In NeurIPS, 2018.

[5] A.V. Do, A. Neumann, F. Neumann, A.M. Sutton. Rigorous runtime analysis of MOEA/D for solving multi-objective minimum weight base problems. In NeurIPS, 2023.

[6] M.A. Hevia Fajardo, P.K. Lehre, S. Lin. Runtime analysis of a co-evolutionary algorithm: Overcoming negative drift in maximin-optimisation. In FOGA, 2023.

[7] P.K. Lehre. Runtime Analysis of Competitive co-Evolutionary Algorithms for Maximin Optimisation of a Bilinear Function. In GECCO, 2022.

[8] T. Lin, C. Jin, M. Jordan. On Gradient Descent Ascent for Nonconvex-Concave Minimax Problems. In ICML, 2020.

[9] A. Maiti, R. Boczar, K. Jamieson, L.J. Ratliff. Query-Efficient Algorithms to Find the Unique Nash Equilibrium in a Two-Player Zero-Sum Matrix Game. arXiv preprint arXiv:2310.16236, 2023.

[10] B. O’Donoghue, T. Lattimore, I. Osband. Matrix Games with Bandit Feedback. In UAI, 2021.

[11] I. Panageas, N. Patris, S. Skoulakis, V. Cevher. Exponential Lower Bounds for Fictitious Play in Potential Games. In NeurIPS, 2023.

[12] T.C. Service, D.R. Tauritz. A No-Free-Lunch Framework for Coevolution. In GECCO, 2008.

[13] D.H. Wolpert, W.G. Macready. No Free Lunch Theorems for Optimization. TEVC, 1997.

[14] D.H. Wolpert, W.G. Macready. Coevolutionary Free Lunches. TEVC, 2005.

---

### Author Response · Authors · 2024-08-14
**Thank-you Message from Anonymous Authors**

Dear Reviewers and Area Chair,

Thank you for your feedback and efforts during the rebuttal phase. We have received the responses from all the reviewers now. We appreciate your time and comments, which have helped us improve our work.

Best Regard, Anonymous Authors

---

### Decision · Program_Chairs · 2024-09-25

**Decision:**

Accept (poster)

**Comment:**

The paper makes a significant theoretical contribution by proving a No Free Lunch (NFL) theorem for black-box adversarial optimization, focusing on Nash Equilibrium (NE). The results are rigorous and provide valuable insights into the limitations of adversarial optimization algorithms. However, concerns were raised about the paper's readability and limited discussion of practical applications. The authors have addressed these concerns in their rebuttal, promising to improve clarity and integrate essential details from the appendix into the main text. Given its theoretical importance and the authors' commitment to revisions, the paper is recommended for acceptance.